# The structural basis of *N*-acyl-α-amino-β-lactone formation catalyzed by a nonribosomal peptide synthetase

Dale F. Kreitler [1], Erin M. Gemmell [2], Jason E. Schaffer[2], Timothy A. Wencewicz [2] & Andrew M. Gulick [1]

Nonribosomal peptide synthetases produce diverse natural products using a multidomain architecture where the growing peptide, attached to an integrated carrier domain, is delivered to neighboring catalytic domains for bond formation and modification. Investigation of these systems can lead to the discovery of new structures, unusual biosynthetic transformations, and to the engineering of catalysts for generating new products. The antimicrobial β-lactone obafluorin is produced nonribosomally from dihydroxybenzoic acid and a β-hydroxy amino acid that cyclizes into the β-lactone during product release. Here we report the structure of the nonribosomal peptide synthetase ObiF1, highlighting the structure of the β-lactone-producing thioesterase domain and an interaction between the C-terminal MbtH-like domain with an upstream adenylation domain. Biochemical assays examine catalytic promiscuity, provide mechanistic insight, and demonstrate utility for generating obafluorin analogs. These results advance our understanding of the structural cycle of nonribosomal peptide synthetases and provide insights into the production of β-lactone natural products.

[1] Department of Structural Biology, Jacobs School of Medicine & Biomedical Sciences at the University at Buffalo, 955 Main Street, Buffalo, NY 14203, USA. [2] Department of Chemistry, Washington University in St. Louis, One Brookings Drive, St. Louis, MO 63130, USA. Correspondence and requests for materials should be addressed to T.A.W. (email: wencewicz@wustl.edu) or to A.M.G. (email: amgulick@buffalo.edu)

β-Lactones are endowed with a highly strained ring system that affords these compounds a unique reactivity among natural products. In particular, β-lactones are well-suited for the inhibition of hydrolases, transferases, ligases, and oxidoreductases[1,2] compared with β-lactams which primarily target penicillin-binding proteins[3]. β-lactones appear in a variety of natural product classes, including terpenoids, polyketides, and nonribosomal peptides[4]. It has been suggested that ~70% of potential biosynthetic gene clusters (BGCs) differ significantly from well characterized BGCs[5–7], providing an impetus for identifying the structural features of enzymes that catalyze the formation of unusual natural products, including those harboring a β-lactone group.

One class of natural products are peptides produced by the nonribosomal peptide synthetase (NRPS) enzymes. NRPSs contain multiple catalytic domains joined in one or more multidomain proteins that operate in an assembly line fashion to produce a peptide from standard and nonproteinogenic amino acids[8]. During the biosynthetic cycle, the nascent peptide is bound as a thioester to a peptidyl-carrier protein domain (PCP, also referred to as a thiolation (T) domain) through a phosphopantetheine (pPant) cofactor and delivered to different catalytic domains for extension and modification[8–10]. The core catalytic domains of NRPSs are the adenylation (A domain) that loads the selected amino acid, the condensation (C domain) that carries out peptide bond formation, and the thioesterase domain (TE domain) that catalyzes release of the mature peptide through hydrolysis or macrocyclization[8].

Structural studies of NRPS modules offer insight into the features that govern the efficient delivery of the substrate to the various catalytic domains. The structures of termination modules SrfA-C[11], EntF[12], and the uncharacterized NRPS AB3403[12], as well as the initiation module of LgrA[13] and the cross-module structure of the DhbF[14], offer clues into the conformational dynamics that drive a three-state structural cycle[10,12]. In state 1, illustrated by the AB3403 protein, the adenylation domain adopts a conformation that catalyzes the initial adenylation of the amino acid substrate[12]. In state 2, a small C-terminal subdomain of the A domain rotates by 140° to accompany the delivery of the downstream PCP domain to the A domain pocket for loading of the pPant cofactor[15]. This thioester-forming conformation has been observed in the structure of EntF[12], as well as DhbF structure[14]. Upon loading of the substrate on the PCP domain, the adenylation C-terminal subdomain reverts to state 1, a conformation in which the loaded substrate is delivered to the C domain for peptide bond formation with an upstream substrate. Simultaneously, the A domain adopts the adenylate-forming conformation to enable activation of another substrate molecule for a subsequent round of extension. Finally, the extended peptide substrate on the PCP is delivered to downstream C or TE in state 3, which has not been observed structurally in the context of a complete module but which can be inferred from existing didomain structures of PCP-TE[16] or of PCP-C[17].

Recent work[18,19] has identified the key steps in the biosynthesis of obafluorin, a nonribosomal peptide from *Pseudomonas fluorescens* (ObiF/D) and *Burkholderia diffusa* (ObiF1/F2/D; Fig. 1). Obafluorin is an *N*-acyl-α-amino-β-lactone with broad spectrum antimicrobial activity[20,21]. Obafluorin is produced by coupling a molecule of 2,3-dihydroxybenzoic acid (2,3-DHB) to the α-amino group of (2*S*,3*R*)-β-hydroxy-*p*-nitro-homophenylalanine (β-OH-*p*-NO₂-homoPhe) via amide bond linkage with accompanying 4-membered lactone formation via the β-hydroxy group. The unusual β-hydroxy-α-amino acid is derived from a crossed aldol reaction of a glycine enolate generated from L-Thr with *p*-nitrophenylacetaldehyde (PNPAA) catalyzed by the PLP-dependent L-Thr transaldolase, ObiH. The NRPS proteins are distributed across two (ObiF/D in *P. fluorescens*) or three (ObiF1/F2/D in *B. diffusa*) polypeptide chains (Supplementary Fig. 1). In *P. fluorescens* the domain orientation of the NRPS module ObiF contains C-A-PCP-TE-MLP-A$_{Ar}$ architecture (MLP, MbtH-like protein, see below). The C-terminal adenylation domain (A$_{Ar}$) activates 2,3-DHB as the corresponding acyl adenylate and loads a free-standing aryl carrier protein (PCP$_{Ar}$) ObiD. The A-domain embedded in ObiF activates β-OH-*p*-NO₂-homoPhe and loads the adjacent PCP domain. The C domain is presumably responsible for amide bond formation while the TE domain catalyzes cyclization to the β-lactone and product release, enabling turnover of the ObiF catalytic cycle. Interestingly, in *B. diffusa*, the A$_{Ar}$ domain for 2,3-DHB is a free-standing protein (ObiF2) that interacts in trans with the multidomain C-A-PCP-TE-MLP NRPS module (ObiF1).

The obafluorin NRPS contains several other surprising features. First, the TE domain, containing a less common catalytic cysteine, is responsible for both β-lactone formation and peptide release. In addition, ObiF contains an integrated MbtH-like protein (MLP), a small domain that has been shown to interact with some adenylation domains to improve protein solubility, or to alter A domain activity or possibly specificity[22–24]. In ObiF from *P. fluorescens*, the MLP is positioned between the TE and terminal A$_{Ar}$ domain. In contrast, the NRPS from *B. diffusa* (ObiF1) terminates at the MLP.

To identify the structural features that govern NRPS-catalyzed obafluorin biosynthesis and to further establish the mechanism outlined previously[18], we present the X-ray crystal structure of *holo*-ObiF1 from *B. diffusa*. The structure shows that the MLP domain interacts with the upstream, non-adjacent adenylation domain, an interaction that to our knowledge has not previously been observed. We identify the catalytic triad in the ObiF1 TE domain that harbors the cysteine residue and also an aspartic acid residue on β-strand 6, a less common position for NRPS thioesterase domains. An in vitro reconstitution assay validates the importance of this MLP interaction, identifies an arginine residue that may be involved in inter-domain acyl transfer from the PCP thioester to the TE domain catalytic nucleophile, and shows that the unusual triad configuration is exploited to enable β-lactone cyclization within the TE domain. Finally, we assessed the substrate scope for the *B. diffusa* enzymes to provide insight into A-domain selectivity and capacity for analog production. The functional and structural studies reported here are consistent with direct β-lactone ring closure catalyzed by the TE domain. Together these results extend our understanding of the structural cycle of the NRPS enzymes and highlight several unanticipated features that should facilitate the identification of more β-lactone-containing natural products.

## Results

**ObiF displays a unique module conformation.** To continue our studies of NRPS systems, we obtained an X-ray crystal structure (3.0 Å) of *holo*-ObiF1 (Fig. 2). Similar to AB3403[12] and SrfA-C[11], and in contrast to the EntF[12,25] and DhbF[14] structures that used an adenosinevinylsulfonimide (AVS) inhibitor to covalently tether the PCP domain pPant arm to the A domain, the ObiF1 structure features a conformation in which the pPant arm of the thiolation domain is docked into the condensation domain cleft. We compared the structure of ObiF1 with previously characterized NRPS enzymes, including large multidomain structures as well as single domain enzymes (Supplementary Table 1). For all domains, the final RMS displacement values are highly dependent on the known conformational changes including the opening and closing of the two lobes of the C domain, the orientation of the C-terminal subdomain of the A

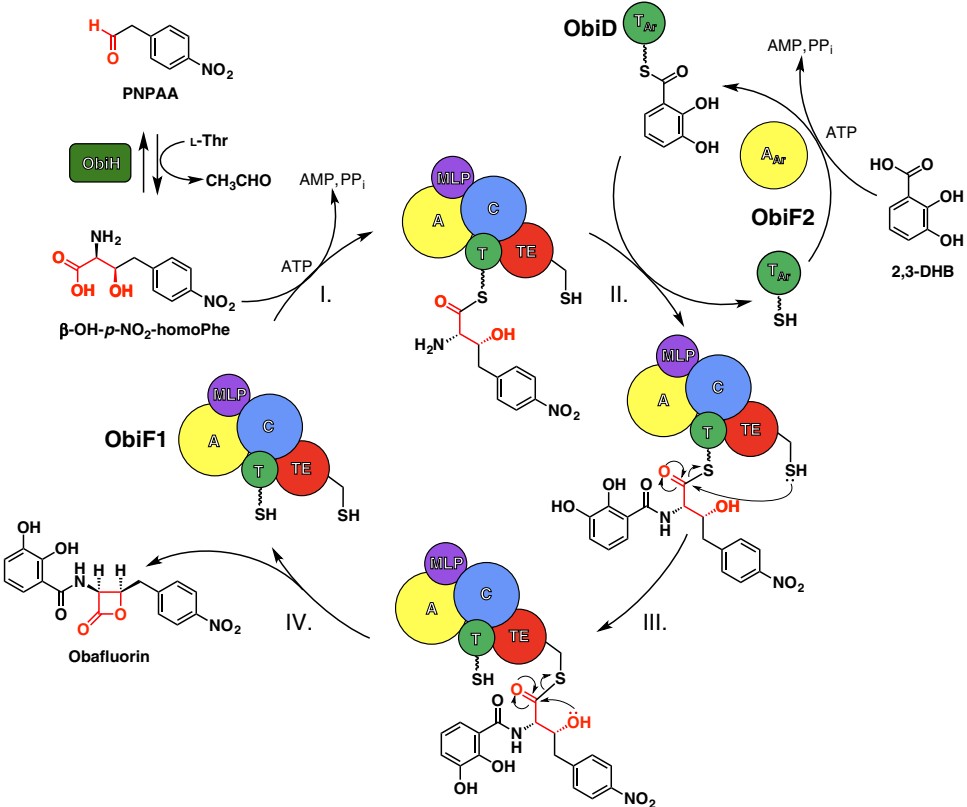

**Fig. 1** Obafluorin biosynthetic pathway. The NRPS catalytic cycle begins with (I) activation and loading of (2S,3R)-β-OH-p-NO₂-homoPhe, generated by *trans*-aldol reaction of p-nitrophenylacetaldehyde (PNPAA) with a glycine enolate catalyzed by the L-Thr transaldolase ObiH, onto the ObiF1 PCP domain (T, green) by the ObiF1 adenylation domain (A, yellow). (II) Concurrently ObiF2 (A_{Ar}, yellow) activates 2,3-dihydroxybenzoic acid (2,3-DHB) and loads it onto ObiD (T_{Ar}, green). (III) Substrate-loaded ObiD then docks onto the ObiF C domain (C, blue), where the α-amino group of β-OH-p-NO₂-homoPhe-T domain thioester is acylated by 2,3-DHB. (IV) Subsequent transthioesterification occurs between the substrate-loaded ObiF PCP domain and TE domain (TE, red) followed by TE domain catalyzed β-lactone formation and obafluorin release. The C-terminal MLP domain (purple) is highlighted to show interactions with the C and A domains of ObiF

domain, and the orientation of the lid in the TE domain. The ObiF1 condensation domain is most similar to calcium-dependent antibiotic (CDA) synthetase, showing a closer approach of the two lobes of the domain. The A domain adopts the adenylate-forming conformation seen in the multidomain AB3403[12] and LgrA[13], as well as isolated domains, such as gramicidin synthetase GrsA[26] and Thr1[27]. The core sheet of the thioesterase domain aligns well with many published structures, showing the highest similarity to the RifR and RedJ thioesterase domains[28,29].

The most striking difference of the ObiF1 structure and previous examples of NRPS modules is that the N-terminal portion of the terminal MLP domain is threaded through the interface formed between the A and C domains to associate with and activate the A domain (Fig. 2d). This domain-spanning region comprises hydrophobic residues buried at the A–C interface in addition to polar and charged residues that make specific contacts to the A or C domain. Salt-bridging interactions occur between Asp1324 (MLP domain) and Arg853 (A domain) and between Arg1317 (MLP domain) and Asp804 (A domain).

**A domain recognition of β-OH-p-NO₂-homoPhe.** The ObiF1 A domain adopts the adenylate-forming conformation (Fig. 3a). As β-hydroxy-α-amino acid activation is a prerequisite for the formation of N-acyl-α-amino-β-lactones, we compared the structure of the ObiF1 embedded A domain with the crystal structure of the L-Thr activating A domain Thr1 (5N9X), which is involved in 4-chlorothreonine biosynthesis in *Streptomyces* sp. OH-5093[27].

The 2S, 3R stereocenters within the β-OH-p-NO₂-homoPhe backbone are set simultaneously during the ObiH-catalyzed *trans*-aldol reaction (Fig. 1)[18]. In both Thr1 and ObiF1, the substrate β-hydroxy does not interact with any protein side chains and is directed into a pocket formed by main-chain atoms. In ObiF1, this cavity is formed by Gly736, Gly737, and Gly764 and the hydroxyl group hydrogen bonds to the carbonyl of Gly764 (Fig. 3c). Thr1 contains glycine residues at these three positions as well. L-Thr, the simplest analog that mimics the β-OH-p-NO₂-homoPhe backbone stereochemistry, is not accepted as a substrate by the ObiF1 A domain[18]. This indicates that the β-hydroxy moiety is not sufficient to confer adenylation activity and additional binding energy provided by the hydrophobic aromatic ring is required.

The specificity-conferring sequence of the ObiF1 A domain is D/A/W/G/C/G/M/I (Fig. 3b), which is similar to that of L-Phe (D/A/W/T/I/A/A/I)[26,30] with a few key differences that accommodate the expanded steric bulk of the p-NO₂-homoPhe side chain relative to that of L-Phe. The specificity-conferring residues of ObiF1 form the hydrophobic pocket for the p-NO₂-phenyl group of the substrate and make no interactions with the polar or charged functional groups along the substrate backbone. Cys734 is located in the pocket and potentially could interact with the NO₂ moiety; however, as currently modeled the thiol is directed away from the substrate. The α-amino group hydrogen bonds to the amide proton of Thr766 and forms a salt-bridging interaction with Asp662. Finally, Ile765 is conserved between both ObiF1 and Thr1 and may play a role in promoting β-strand dihedral angles

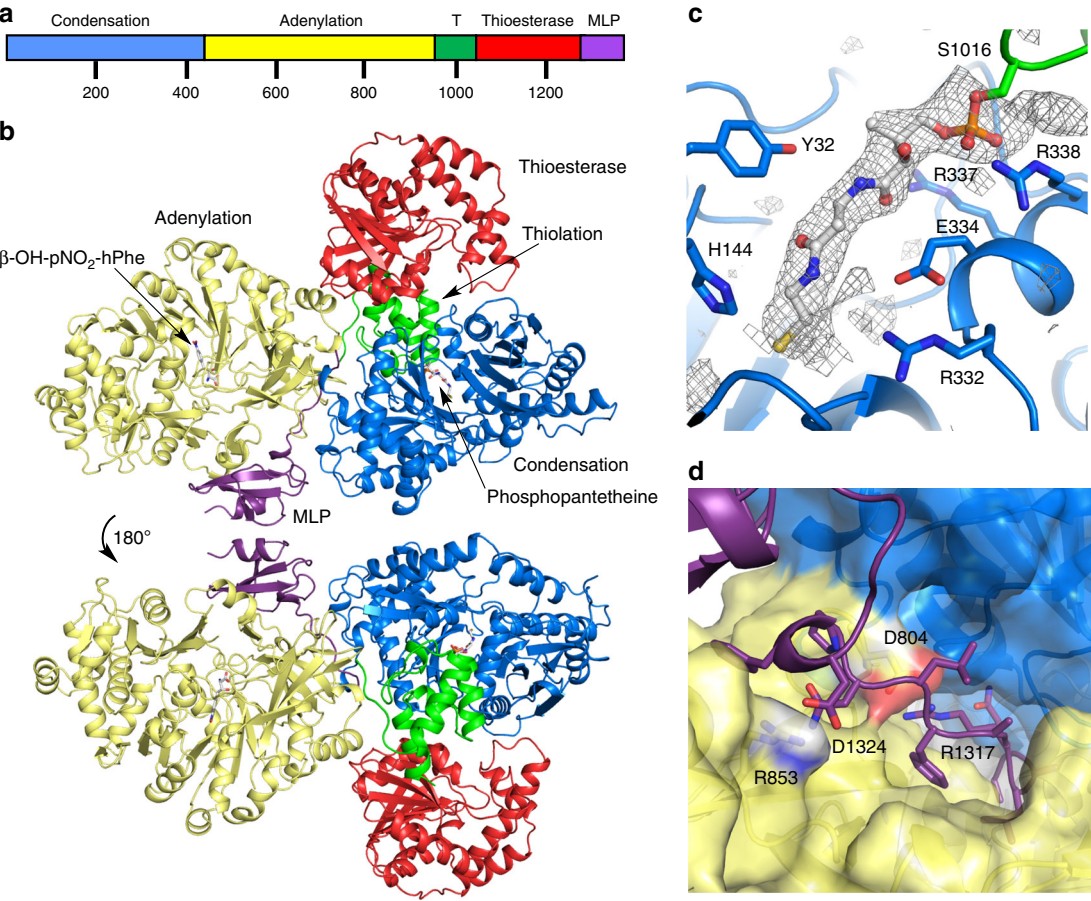

**Fig. 2** Structure of the BD-ObiF1 module. **a** ObiF1 from Burkholderia diffusa possesses a C-A-PCP-TE-MLP domain architecture. **b** Front and back side view of the holo-ObiF1 module. **c** The pPant arm is positioned in the C domain near the canonical catalytic C domain His residue (His144). Electron density (contoured at 3.0σ) calculated with $mF_o-DF_c$ coefficients from an omit map (ligand occupancy set at zero) generated with simulated annealing. **d** A surface representation of the A and C domain interface illustrates the salt-bridging interactions that occur between the N-terminal region of the MLP domain and the A domain

that orient the Gly764 carbonyl toward the β-OH-*p*-NO₂-homoPhe β-hydroxyl group, thus promoting H-bond formation.

**The C-terminal MLP domain is critical for activity**. The C-terminal MLP domain in ObiF1 interacts with the upstream, embedded A domain. MLPs bind to A domains and play important roles in some NRPS modules by enhancing A domain activity, improving module solubility, or both[22,23]. While many MLPs are expressed as free-standing proteins, some proteins, including SlgN1[31] and NikP1[32], possess an MLP that is appended to the N-terminus of its partner A domain. With SlgN1 or NikP1 serving as a model, the presence of the MLP domain in the *P. fluorescens* ObiF protein between the TE domain and the C-terminal $A_{Ar}$ domain suggested a similar interaction with the immediate downstream domain. With the separate ObiF1 and ObiF2 of *B. diffusa*, this domain architecture raised the possibility that the MLP might interact in trans with the free-standing ObiF2 $A_{Ar}$ domain. Surprisingly, the C-terminal MLP binds to the upstream A domain of ObiF1 (Fig. 3a) through the archetypical MLP-A domain interaction. This interaction motif comprises two MLP tryptophan residues (Trp1343 and Trp1353) that form a pocket around an A domain alanine (Ala841; Fig. 3d). This critical interaction has been observed in all crystal structures of MLP domains in complex with an A domain, regardless of whether the A domain is free-standing or in the context of a module[25,31]. The MLP consensus sequence is NXEXQXSXWPX₅PXGWX₁₂

LX₇WTDXRP, where X represents any amino acid[33]. The C-terminal WTDXRP motif is missing from the ObiF1 MLP domain (Supplementary Fig. 2) perhaps an indication, as previously suggested[33], that fused MLPs may have relaxed constraints due to intramolecular interaction.

The presence of the C-terminal MLP should influence the structure of the ObiF1 module. In other NRPS termination modules, AB3403, EntF, EntF/MLP, and SrfA-C[11,12,25], the TE domain adopts multiple positions as it is tethered to the core domains only through a single linker to the PCP, which itself can adopt multiple positions as it migrates between different domains. The ObiF1 structure is unique as the C-terminal MLP domain serves to 'anchor' the TE domain to the core of the module. The relative position of the domains in ObiF1 is most similar to that of the SrfA-C structure in which the TE domain rests on top of the C domain.

We probed this MLP-adenylation interaction in several experiments. First, adopting the strategy used with SlgN1[31], the alanine residue that inserts into the Trp pocket was mutated to glutamate (ObiF1 A841E) to disrupt this interaction. We also generated a truncated construct (ObiF1-del1304) in which the protein was terminated after the TE domain. We expressed this protein alone, or with a tandem expression system to co-express the ObiF1-del1304 protein with the ObiF1 MLP domain, residues 1304 - 1390 in trans (a construct called ObiF1 rbs1303/4). Upon purification the yield of these three protein constructs, A841E, ObiF1-delMLP, and rbs1303/4, was much lower than WT

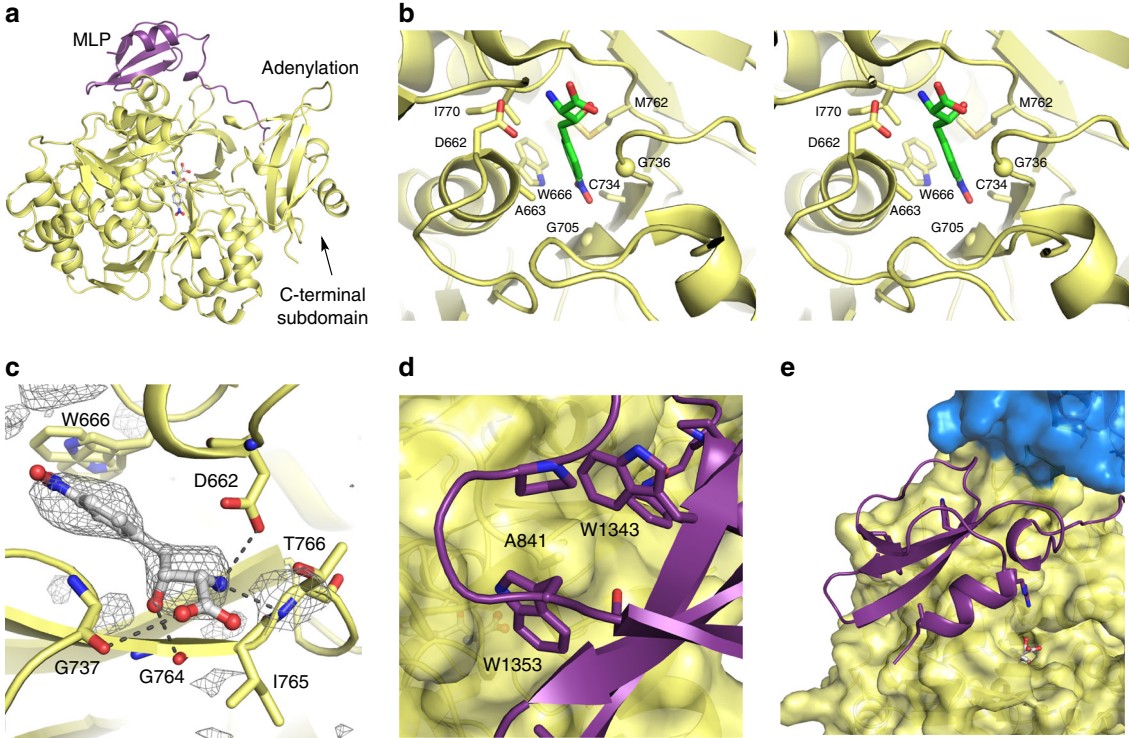

**Fig. 3** ObiF1 Adenylation domain structure and MLP interaction. **a** The MLP domain (purple) interacts with the A domain (yellow) in the adenylate-forming conformation. **b** Stereorepresentation of the specificity determining residues of the substrate binding pocket of the adenylation domain. Cα positions of two glycine residues are indicated with a sphere. Only a single conformation of the side chain of Trp666 is shown. **c** β-OH-p-NO$_2$-homoPhe binds in the A domain active site where the β-hydroxyl forms H-bonds with backbone amide carbonyls of Gly737 and Gly764. Electron density (contoured at 2.5σ) calculated with mF$_o$–DF$_c$ coefficients from an omit map (ligand occupancy set at zero) generated with simulated annealing. The α-amino group forms a salt-bridging interaction with Asp662 and an H-bond with the amide proton of Thr766. The side chain of Trp666, which adopts two alternate conformations, contributes to the hydrophobic pocket around the p-NO$_2$-phenyl ring. **d** Trp1343 and Trp1353 of the MLP domain form a pocket around Ala841 of the A domain. **e** The MLP domain in relation to the C domain (blue) and A domain active site

ObiF1. Analysis of the final size exclusion chromatography purification steps demonstrated that the three proteins with missing or disrupted MLP interaction were much more likely to aggregate and elute in the column void volume (Supplementary Fig. 3). We also produced the ObiF1 MLP domain recombinantly and the ObiF1 proteins were evaluated functionally in the presence or absence of exogenous ObiF1 MLP.

We developed a simple in vitro reconstitution assay for ObiF1/F2/D/H using LCMS to quantify product ion counts (Fig. 4; Supplementary Fig. 4). The advantage of this approach is the ability to distinguish turnover pathways for the catalytic cycle by monitoring product formation as opposed to substrate consumption, which does not always correlate with actual product formation. Starting from phenylacetaldehyde (PAA), the L-Thr transaldolase ObiH generates a steady equilibrium concentration of β-OH-homoPhe that is loaded as the ObiF1 PCP thioester and condensed with the ObiF2-generated 2,3-DHB-ObiD thioester to form the final N-(2,3-OH-benzoyl)-β-OH-homoPhe PCP thioester. There are three mechanistic possibilities for product cleavage from the final PCP thioester intermediate, including solvolysis by glutathione (GSH) in the reaction buffer (path a), acyl transfer to the TE domain followed by beta-lactone formation (path b), or hydrolysis (path c) (Fig. 4a). Any β-lactone formed is immediately trapped as the GSH-thioester under these assay conditions. We validated the use of thiols as β-lactone trapping agents using obafluorin β-lactone and the corresponding β-hydroxy acid purified from *P. fluorescens* ATCC 39502 culture supernatant (Supplementary Figs. 5, 6, 7)[18]. Formation of thioester was instantaneous and quantitative from the corresponding β-lactone.

In addition, the GSH-thioester has greater ionization potential and increased stability ($t_{1/2}$ ~ 3 h at pH 7) compared with the parent β-lactone ($t_{1/2}$ ~ 5 min at pH 7), making thioesters a useful proxy for quantifying β-lactone formation[18].

Product formation by 1 μM WT ObiF1 was not affected by the presence of exogenous MLP domain at 1, 10, or 25 μM (Fig. 4b). This is consistent with occupation of the MLP-binding site by the tethered C-terminal MLP domain in the native ObiF1 module. The truncated ObiF1 mutant del1304 at 1 μM generated ~50-fold lower product ion counts, while addition of exogenous MLP at 1 μM restored product ion counts to WT levels. Since ObiF1-del1304 was expressed and purified from *E. coli* deficient in native MLP production, this implies that the NRPS is still functional, albeit at reduced efficiency, without an MLP interaction. Increasing MLP concentrations to 10 or 25 μM did not increase product ion counts suggesting saturation of the MLP-binding site at stoichiometric levels of added MLP relative to ObiF1-del1304. Coexpression of the ObiF1 MLP domain with the del1304 mutant from the same pET28 plasmid produced protein sample rbs1303/4 that gave enhanced product ion counts relative the del1304 mutant alone, but still at approximately fourfold reduced levels to WT. Similar to the del1304 mutant, addition of exogenous MLP recovered product ion counts for the rbs1303/4 protein reaction to WT levels. Collectively, the results from MLP supplementation of del1304 and rbs1303/4 support the notion that in trans MLP interactions compensate for deletion of the C-terminal MLP domain in ObiF1.

Similar to the del1304 mutant, the A841E mutant generated ~25-fold lower product ion counts compared with WT

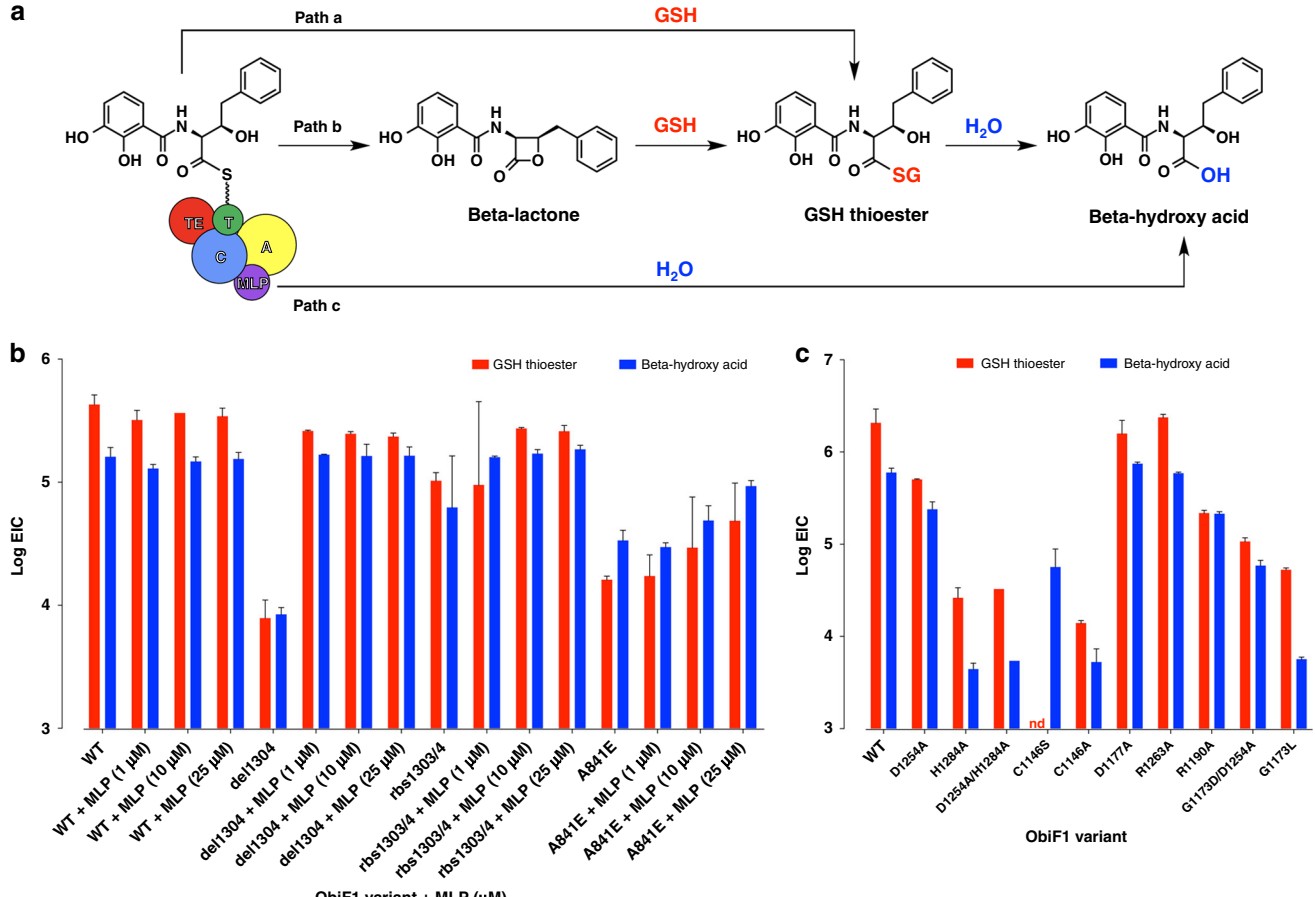

**Fig. 4** Functional analysis of MLP-dependence and triad residues. **a** Three mechanistic pathways for product release from the penultimate N-(2,3-OH-benzoyl)-β-OH-homoPhe PCP thioester generated using an ObiF1/F2/H/D functional assay. GSH was included as a trapping agent to enable quantification of product ion counts by LCMS. Bar graphs in (**b**) and (**c**) represent extracted ion counts (EICs) for the GSH-thioester ($m/z = 621$ for $[M + H]^+$ ion) and β-hydroxy acid ($m/z = 332$ for $[M + H]^+$ ion) normalized to a phenylalanine internal standard at the 3-h reaction time point. The y-axis represents the log (EICs) and the x-axis denotes the ObiF1 variant used in the enzymatic reaction along with concentration of exogenous MLP (if added). Error bars represent standard deviations for at least two independent trials. Bar graphs for additional time points (0.5 and 24 h) are provided in the Supplementary Information. nd = not detected by LCMS

ObiF1 (Fig. 4b). Contrary to del1304, however, addition of 1 μM MLP provided only a modest enhancement in product ion counts. Increasing the concentration of added MLP to 10 and 25 μM further enhanced product ion counts without ever reaching WT product levels (still approximately eightfold lower than WT at 25 μM MLP). The apparent dose dependence of MLP-promoted product formation for the A841E mutant is consistent with a reduced binding affinity of the MLP domain at the modified MLP-binding site. To gain insight on the effect of MLP on ObiF2 activity, we checked levels of the 2,3-DHB acyl adenylate by LCMS for reactions with WT, del1304, rbs1303/4, and A841E ObiF1 variants (Supplementary Fig. 8). For all proteins, the 2,3-DHB adenylate levels reached equilibrium and stayed constant at 30 min, 3 h, and 24 h consistent with MLP-independent activity for the $A_{Ar}$ ObiF2. The MLP-related mutant activities in the functional assay support interaction of the MLP domain with the ObiF1 module as observed in the crystal structure. However, it does appear that tethering the MLP domain to the NRPS module is not required suggesting the MLP domains acting *in trans* for related NRPSs may have interactions with upstream A domains embedded in NRPS modules.

**ObiF TE domain adopts a RifR-like triad configuration.** The ObiF TE domain (*P. fluorescens*) catalytic cysteine is required for

β-lactone formation[18]. We observed a hydrogen bonding network of the catalytic triad at residues Cys1146, His1284, and Asp1254 (Fig. 5). Markedly, the aspartic acid residue is repositioned compared with its location in the AB3403 and EntF TE domains, or in the recently characterized TE domain of valinomycin NRPS system, which was structurally characterized by replacing the catalytic serine with a non-canonical amino acid 2,3-diamino-propionate to capture a stable ester mimic[34]. The aspartic acid residue is shifted away from the catalytic cysteine residue and occurs on β-strand 6 rather than β-strand 5, as in the EntF or AB3403 TE domains. The resultant triad configuration is similar to that of RifR (3FLB; rifamycin biosynthesis), a type II TE domain with acyl-PCP hydrolase activity (Fig. 5d)[28,29]. On β-strand 5, where the aspartic acid residue occurs in EntF and AB3403, ObiF1 displays a glycine residue (Gly1173), possibly to minimize steric clash with the substrate-loaded pPant arm during acyl transfer to Cys1146 of the TE domain active site.

After the PCP domain is successfully loaded with the intermediate N-acyl β-hydroxy-α-amino acid, the putative mechanism for ObiF1-catalyzed β-lactone formation requires transthioesterification between the substrate-loaded pPant arm and the Cys1146 thiol, where the presumed oxyanion intermediate is partially stabilized by His1145, followed by 4-*exo-trig* ring closure in which the β-hydroxy group attacks the

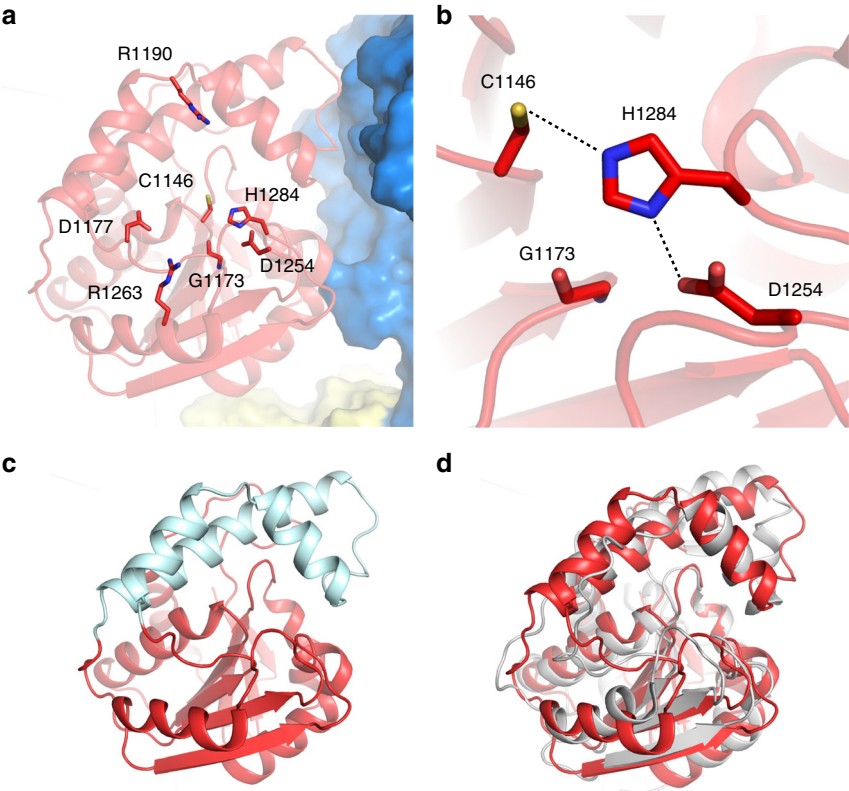

**Fig. 5** Structure of ObiF1 TE domain. **a** The ObiF1 TE domain (red) is adjacent to the C domain (blue); active site residues that were targeted for mutation are highlighted. **b** Catalytic triad residues Cys1146, His1284, and Asp1254. In ObiF1, Gly1173 occurs at the β-strand 5 position where the aspartic acid residue is typically found in the His-Asp dyad of EntF-like TE domains. **c** The ObiF1 TE domain (red) contains a three-helix lid-loop region (cyan). **d** ObiF1 TE domain (red) is superimposed with the RifR type II TE domain (3FLB; gray) using PyMol

substrate-loaded Cys1146 thioester (Supplementary Fig. 9). To establish the identity of the catalytic triad residues we mutated the His-Asp-Cys triad (H1284A, D1254A, H1284A/D1254A, C1146S, C1146A). These mutant proteins were reconstituted in vitro and evaluated for product formation in the reconstitution assay monitored by LCMS. Each mutant displayed a diminished capacity for product formation relative to WT ObiF1 (Fig. 4c). Product ion counts from the D1254A mutant reaction reduced by approximately eightfold compared with WT ObiF1, whereas product formation by the H1284A mutant and the D1254/H1284A double mutant were more severely compromised at ~80-fold reduction in product ion counts. These product ion levels were similar to the C1146A mutant, which stalls the NRPS cycle at the penultimate *N*-(2,3-OH-benzoyl)-β-OH-homoPhe PCP thioester blocking transfer to the TE domain and leaves only solvolysis pathways a and c outlined in Fig. 4a to turn over the NRPS catalytic cycle through release of GSH-thioester or β-hydroxy acid, respectively. Both processes are slower than TE-mediated product release in WT ObiF1. As expected, the C1146S mutant exclusively generated product ion counts for the β-hydroxy acid, supporting the notion that transfer to the TE domain was achieved leading to a TE S1146 oxoester and product release via water hydrolysis and not GSH solvolysis, which requires a transthioesterification pathway to be thermodynamically possible. The similarities in quantity and ratio of GSH-thioester and β-hydroxy acid product ion counts suggest that the D1254A/H1284A double mutant is dominated by the H1284A mutation, which might block efficient transfer of substrate to the TE domain similar to the C1146A mutant.

To examine the apparent preference of ObiF1 for a RifR-like over an EntF-like triad configuration we generated a ObiF1 double

mutant with an EntF-like positioning of the aspartic acid residue (G1173D/D1254A). This double mutant displayed a reduced activity relative to the D1254A single mutant in the ObiF1/F2/D/H reconstitution assay, as evidenced by an approximately fivefold reduction in product ion counts (Fig. 4c). The attenuation of the G1173D/D1254A double mutant activity relative to the D1254A single mutant activity implies the existence of an additional factor that can impact substrate loading of Cys1146, beyond reduced basicity of the histidine imidazole. We hypothesized that the reduced activity of the G1173D/D1254A double mutant was attributable to steric hindrance blocking access to the Cys1146 thiol during thioester exchange from the PCP. For successful transthioesterification to occur between the pPant arm and the Cys1146 thiol, the TE domain lid (Fig. 5c) must open partially and allow the substrate-loaded pPant arm to slide into the TE domain active site. An amino acid side chain at residue 1173 may occlude the pPant arm from the Cys1146 thiol nucleophile, thus slowing the rate of transthioesterification. To test this steric occlusion hypothesis, and to exclude sub-optimal positioning of the Asp1173 side chain in the G1173D/D1254A double mutant as a factor contributing to slower transthioesterification, we tested a G1173L mutant in the ObiF1/F2/D/H reconstitution assay (Fig. 4c). This mutant, which maintains the wild-type RifR-like triad configuration, showed even greater reduction in product ions (up to ~100-fold) in activity compared with the G1173D/D1254A double mutant. Thus, we conclude that an EntF-like configuration of the TE domain triad results in steric inhibition of the Cys1146 thiolate and, therefore, this type of triad configuration is disfavored in ObiF1.

To explore other residues that might be involved in substrate transfer from the PCP to TE domains, we prepared Ala mutants

of Asp1177, Arg1263, and Arg1190 residues around the ObiF1 TE active site. The D1177A and R1263A mutants generated product ion counts equivalent to WT ObiF1, while the R1190A mutation resulted in a ~10-fold reduction in product ions. Product levels for the R1190A mutant were still ~16-fold higher than the C1146A mutant suggesting that substrate transfer to the TE domain is still possible at reduced efficiency relative to WT ObiF1. We note that we employed phenylacetaldehyde (PAA) in the ObiF1/F2/D/H reconstitution assay in both WT control and mutant reactions; therefore, any potential ionic interactions between the Arg1190 side chain and the obafluorin nitro group can be excluded as a contributing factor in substrate pre-organization.

**The substrate scope of ObiF1**. To probe the substrate scope of the NRPS module we evaluated a series of aldehydes (**1a–h**) and benzoic acid derivatives (**3a–u**) in the ObiF1/F2/D/H reconstitution assay (Fig. 6, Supplementary Fig. 10). For reactions with variable aldehyde substrates, the benzoic acid substrate was always 2,3-DHB (**3a**). For reactions with variable benzoic acid substrates, the aldehyde substrate was always PAA (**1a**). Product nomenclature reflects the identity of the aldehyde (first letter) and benzoic acid (second letter) substrates. We again used the thiol capture approach for quantifying β-lactones, this time using dithiothreitol (DTT) as the reactive thiol nucleophile. The use of DTT resulted in initial formation of the DTT-thioester that rearranges to the more thermodynamically stable DTT-oxoester that does not undergo further hydrolysis under the reaction conditions (Supplementary Figs. 5–7). This was a useful trapping agent for distinguishing the diverse product ions for the substrate screen given there are multiple peaks, some interconverting, to correlate during reaction monitoring by LCMS. In addition, the DTT acyl transfer rearrangements confirmed that thioesters are the kinetic products and oxoesters are thermodynamic products, which is consistent with our logic for distinguishing product release pathways in the ObiF1 mutant reactions. We used LCMS to quantify product ion counts corresponding to β-hydroxy amino acids (**2**), thioesters (**5**), and *N*-acyl-β-hydroxy acids (**6**) that were normalized to a phenylalanine internal standard. All product identities were confirmed using high-resolution mass spectrometry (Supplementary Data 1).

The aldehydes (**1a–h**) are converted to the corresponding β-hydroxy-α-amino acids (**2a–h**) by transaldolase ObiH to provide a steady equilibrium concentration of the NRPS substrate that can only be driven toward *N*-acylated products by ATP consumption. In most cases, the β-hydroxy-α-amino acids (**2a–h**) can be detected by LCMS (Fig. 6b) indicating that ObiH is broad in substrate scope. However, it is noteworthy that even in some cases where β-hydroxy-α-amino acids (**2**) are not detectable, we still observed product formation (e.g., **5ha, 6ha**). There are also instances where the amino acid substrate was detected at early time points and was fully consumed over the course of 24 h, presumably due to complete conversion of the starting aldehyde (the limiting reagent) to thioester (**5**) and β-hydroxy acid (**6**) products (e.g., **5ba, 6ba**). The best substrates for ObiF1 were β-hydroxy-α-amino acids (**2a, 2b**) derived from phenylacetaldehydes that most closely resemble the natural PNPAA aldehyde substrate that gives rise to *p*-nitro-substituted obafluorin (Fig. 1). Benzaldehydes were converted to the corresponding β-hydroxy-α-amino acids (**2c, 2d**) at low levels, but no NRPS products were detected indicating that the ObiF1 A domain poorly activates these substrates. Products resulting from linear alkyl (**5ea, 6ea**) and extended aryl (**5fa, 6fa, 5ha**, and **6ha**) substrates were also generated. This indicates that the phenyl ring or a hydrophobic alkyl chain must extend deep into the ObiF1 A

domain substrate binding pocket in order to avoid unfavorable steric clashes, as indicated by the crystal structure showing active site occupation by the natural substrate β-OH-*p*-NO$_2$-homoPhe (Fig. 3b). This is consistent with our previous rationalization for exclusion of L-Thr and L-Ser, naturally occurring β-hydroxy-α-amino acids, as substrates. It is worth mentioning that while a phenethyl side chain was converted to product (**5fa, 6fa**) the corresponding (*E*)-phenethylenyl side chain was not activated by ObiF1 indicating that substrate flexibility plays a role in A domain binding and/or activation.

The A$_{Ar}$ domain, ObiF2, was also capable of activating a wide range of benzoic acid substrates (Fig. 6c). The natural substrate, 2,3-DHB (**3a**), gave the highest product ion counts (**5aa, 6aa**) and a clear preference for 2,3-substitution of the benzoic acid derivative emerged when comparing product ion counts for the 21 substrates that were included in the study. For example, 2-OH and 3-OH benzoic acids (**3c** and **3d**) were both activated and converted efficiently to products **5ac/6ac** and **5ad/6ad**, respectively; while 4-OH benzoic acid gave significantly lower product ion counts only detectable for the thioester **5ae**. Some halogen substituents (Cl and F) at the 2,3-positions were accepted as substrates, but this appeared to be somewhat dependent on atomic radius. Larger halogens, Cl and Br, at the 4-positions gave no product ion counts and 2,3-Cl (**an**) products were not detected while 2,3-F (**aq**) product ions were strong. We checked benzoic acid acyl adenylate levels by LCMS and found 100% correlation between the presence of acyl adenylate and the presence of product ions in the enzymatic reactions (**aa–ad, af, ag, al, am, ao–aq**) (Supplementary Fig. 11). In addition, the absence of acyl adenylate detection by LCMS correlated 100% to absence of product ion counts (**ah–aj, an, ar, at, au**). However, there were some instances where acyl adenylate was detected, but no product ions were formed (**ae, ak**) and the amount of acyl adenylate ions did not always correlate with the amount of product ions, which is to be expected since each analog will have variable stability and ionization potential under the assay conditions. This implies that product formation and scope of analog production is determined by the adenylation selectivity of ObiF2 for benzoic acids and the domain specificity in the NRPS module ObiF1 for structurally modified intermediates.

## Discussion

Here we describe the structural basis that defines the genetic signatures of *N*-acyl-α-amino-β-lactone formation from the ObiF1 NRPS system. ObiF1 has a unique MLP domain that forms a non-local protein-protein interaction with the upstream A domain. The ObiF1 A domain tolerates a variety of bulky hydrophobic β-hydroxy-α-amino acids while excluding activation of the proteinogenic β-hydroxy-α-amino acids L-Ser and L-Thr. ObiF1 is somewhat unusual in that its resulting β-lactone is the *cis* stereoisomer, a consequence of the β-OH-*p*-NO$_2$-homoPhe backbone stereochemistry which is set during the ObiH-catalyzed cross-aldol reaction[4,18]. The ObiF1 A domain is likely constrained to activate β-hydroxy-α-amino with the same stereochemistry as β-OH-*p*-NO$_2$-homoPhe. β-Hydroxy-α-amino acids with configurations similar to that of *allo*-L-Thr would likely be unable to participate in the favorable backbone carbonyl H-bond network observed in the A domain active site and thus disfavored as an ObiF1 substrate.

We observed an unexpected interaction of the C-terminal MLP domain with the upstream A domain in the ObiF1 protein. This interaction was validated biochemically and appears to be a new type of interaction. While it has previously been demonstrated that free-standing MLPs can interact with multiple NRPS A domains within a biosynthetic pathway[35–37], this observation

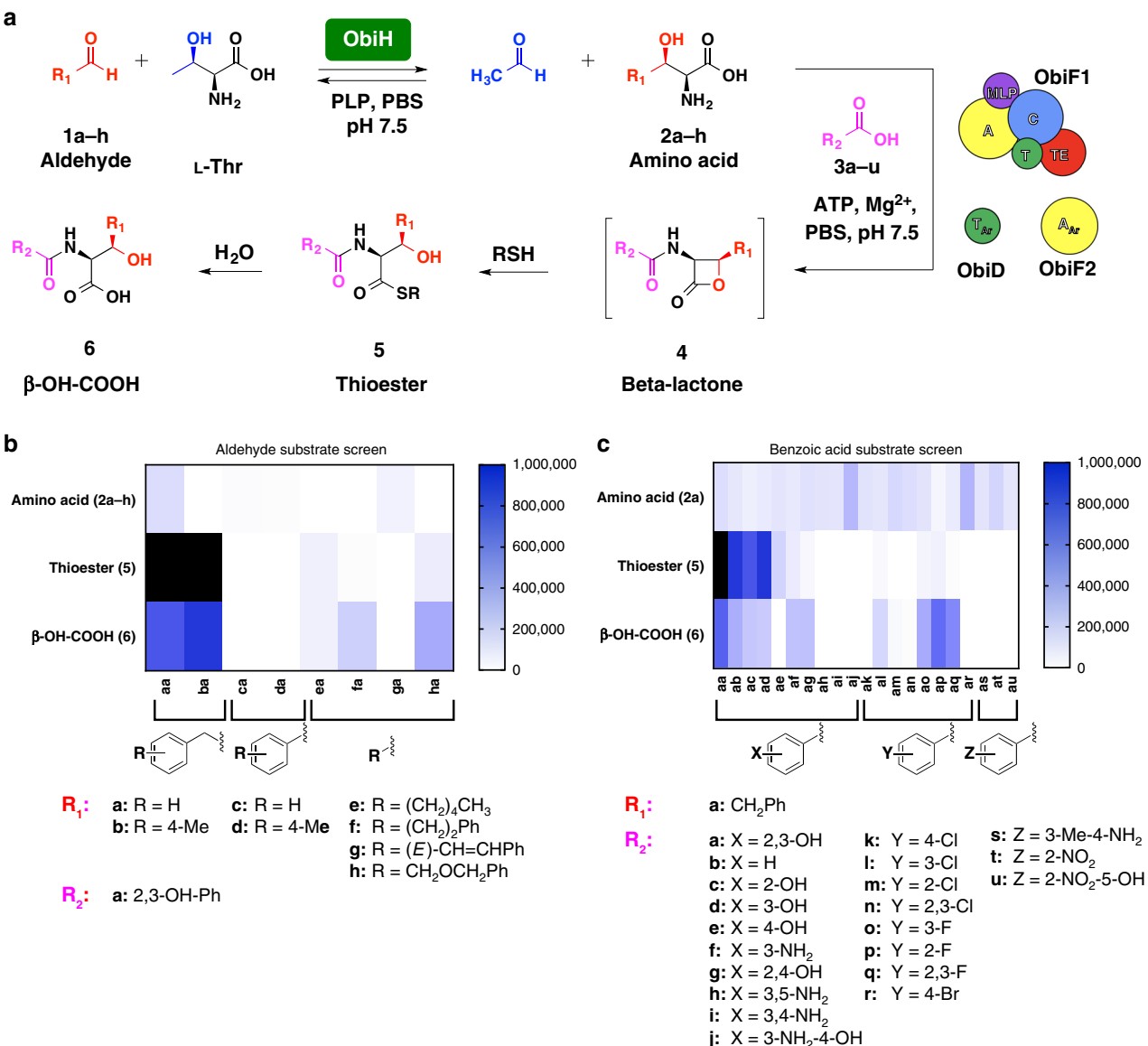

**Fig. 6** Probing NRPS substrate scope. **a** Scheme depicting the reaction conditions for the coupled ObiF1/F2/D/H in vitro enzyme assay using a thiol kinetic capture strategy for the β-lactone product. DTT was employed as the reactive thiol. Thioester and β-hydroxy acid product ion counts were quantified via LCMS (see Supplementary Data 1 for [M + H]⁺ m/z values and EIC traces). Heat maps represent the extracted ion counts for the amino acid (**2a–h**), thioester (**5**), and β-hydroxy acid (**6**) resulting from the use of variable **b** aldehyde (**1a–h**) or **c** benzoic acid (**3a–u**) substrates normalized to a phenylalanine internal standard for one trial at the 24 h reaction time point. For compounds (**4**–**6**), the first letter in the two-letter nomenclature, (**a–h**), represents the structure of R₁ while the second letter in the two-letter nomenclature, (**a–u**), represents the structure of R₂. Heat maps for additional time points (0.5 and 3 h) are provided in the Supplementary Information. The color scale to the right of each figure reflects peak counts; peaks above 1,000,000 counts are colored black

illustrates that internal MLPs can also interact beyond an immediately downstream A domain. Like other MLPs, the domain from ObiF1 enhances protein stability and function, even in light of the new structural constraints that it adds to the modular conformation.

The GXCXG TE domain motif appears to facilitate β-lactone formation via two related processes which include acyl transfer (energy-neutral transthioesterification in this case) to the active site cysteine thiolate followed by 4-*exo-trig* ring closure to generate the β-lactone. Acyl transfer to the TE domain ensures that the thioester is protected from solvolysis and enters the TE domain active site environment that promotes kinetically favorable β-lactone ring formation, probably with conformational control leading to stabilization of the *4-exo-trig* cyclization

transition state. This dominant hydrolysis of the C1146S mutation results from the enhanced reactivity of a thioester versus an oxoester with a hydroxyl nucleophile, which was also noted in the increased promiscuity of an engineered TE domain from the pikromycin polyketide synthase system[38]. The cysteine thiolate is generated by deprotonation of the Cys1146 thiol via His1284 and Asp1254, and removal of either side chain significantly diminishes ObiF1 activity. Surprisingly, the ObiF1 TE domain positions the catalytic Asp on β-strand 6 in a configuration that is similar to the type II TE domains RifR and RedJ. The EntF-like triad configuration (Asp on β-strand 5) sterically inhibits substrate loading onto Cys1146. Acyl transfer from the pPant arm to the TE domain cysteine competes with solvolysis by environmental thiols and water; perhaps the larger lid-loop region of the ObiF

TE domain and the RifR-like triad configuration allow less solvent into the active site compared with TE domains in which the only function is substrate release. Both EntF and ObiF1 NRPSs generate N-2,3-DHB-β-hydroxy-α-amino acid intermediates that differ only in the identity of the β-hydroxy-α-amino acid substrate, L-Ser and β-OH-p-NO$_2$-homoPhe, respectively. The similarity in NRPS domain orientation between ObiF1 (C-A-PCP-TE-MLP) and EntF (C-A-PCP-TE) highlights the importance of the alternate catalytic properties of NRPS TE domains in determining product outcome from the thioester tethered N-2,3-DHB-β-hydroxy-α-amino acid intermediates, iterative trimerization and macrolactonization for EntF and β-lactone formation for ObiF1.

The ObiF1 TE domain possesses some general features that might facilitate β-lactone formation. These features include a large lid loop that occludes solvent from the active site Cys1146 residue and a catalytic triad configuration similar to that observed in the type II TE domains RifR and RedJ. Presumably, this configuration allows adequate space for a hydrophobic pocket to form adjacent to the Cys1146 side chain. This pocket reduces the conformational degrees of freedom available to the tethered substrate, thus promoting cyclization and subsequent product release. Expansion of this active site pocket via mutation may further expand the substrate scope of the ObiF TE domain.

Our structural and functional characterization of the ObiF1/F2 NRPS assembly line from *B. diffusa* provides an improved understanding of NRPS domain interactions, substrate selectivity, and termination chemistry. We have established the structural blueprint for a β-lactone forming NRPS facilitated by a unique TE catalytic domain that kinetically favors formation of the strained 4-membered ring product. Our findings can be applied toward genome mining efforts, chemoenzymatic syntheses, and mechanistic studies of related NRPS and PKS biosynthetic assembly lines that generate strained β-lactone and β-lactam 4-membered heterocycles.

## Methods

**Purification of ObiF1 for crystallographic analysis**. The *Burkholderia diffusa* gene encoding for ObiF1 (ObiF1; KUZ09184.1) was synthesized, codon optimized (Supplementary Fig. 12), and cloned into a pET28 vector (NdeI/HindIII) encoding for an N-terminal hexahistidine tag (pET28-ObiF1; Genscript; Piscataway, NJ, USA). In total, 1.5 L of TB media (30 μg/mL kanamycin) was inoculated with 1.5 mL of an LB (30 μg/mL kanamycin) overnight culture (37 °C, 250 rpm) of *E. coli* BL21(DE3) cells harboring the pET28-ObiF1 plasmid. The 1.5 L TB culture was grown to an OD$_{600}$ of ~0.7 (37 °C, 250 rpm) at which time IPTG (500 μM) was added and the culture was incubated for 18 h (15 °C, 250 rpm). Cells were harvested via centrifugation (5000 × *g*, 35 min, 4 °C) and flash frozen in liquid nitrogen. Subsequent purification steps were performed at 4 °C. Cells were re-suspended in lysis buffer (50 mM HEPES pH 8.0, 500 mM NaCl, 0.2 mM TCEP, 20 mM imidazole, 10% v/v glycerol; 20 mL lysis buffer per gram cell pellet).

Cells were lysed by mechanical disruption (Branson Sonifier), and the resulting cell lysate was clarified via ultracentrifugation (40,000 rpm, 40 min, 45-Ti rotor), filtered (0.45 μm), and passed over a Ni-loaded HisTrap column (2 × 5 mL columns in series). The column was washed with 2.5 column volumes of 10% v/v elution buffer (50 mM HEPES pH 8.0, 500 mM NaCl, 0.2 mM TCEP, 300 mM imidazole, 10% v/v glycerol) in lysis buffer and immobilized protein was eluted with 3 column volumes of elution buffer. Protein-laden fractions were pooled (11.9 mg/mL protein) and combined with the promiscuous phosphopantetheinyl transferase Sfp (400 nM), MgCl$_2$ (1 mM), and five equivalents of coenzyme A (400 μM). The pantetheinylation reaction was then rocked gently for 3 h at 4 °C after which the reaction was further purified by gel filtration (GE Healthcare HiLoad 16/600 Superdex 200, 0.8 mL/min) in 50 mM HEPES pH 8.0, 500 mM NaCl, 0.2 mM TCEP, 10% v/v glycerol. Finally, ObiF1 protein was exchanged into a crystallization buffer composed of 25 mM HEPES pH 8.0, 25 mM NaCl, 0.4 mM TCEP, 5% v/v glycerol (15 mL Amicon Centrifugation filter, 50 kDa MWCO). Concentrated ObiF1 protein solution (36.5 mg/mL) was flash frozen in liquid nitrogen and stored at −80 °C (Supplementary Fig. 13).

**Crystallization of ObiF1**. Initial crystallization conditions were identified through the Hauptman-Woodward Medical Research Institute High-throughput Screening (HTS) Laboratory. Preliminary crystals, used to obtain an initial set of phases, were grown using the microbatch-under-oil technique; crystallization droplets were

setup under paraffin oil in an untreated, polystyrene 72 well microbatch plate from Hampton Research (Aliso Viejo, CA, USA) at 23 °C. Crystallization droplets (2 μL) were composed of a 1:1 ratio of ObiF1 protein solution and a crystallization cocktail (90 mM HEPES pH 6.8, 5% w/v 1,3-dimethylimidazolium dimethyl phosphate, 27% w/v PEG3350).

These preliminary crystals were cryo-protected by treatment with mixtures of the crystallization cocktail, an aqueous sucrose solution (50% w/v), and cryo-solution A using the following procedure. Cryo-solution A was made by combining ethylene glycol (neat) and an aqueous solution of non-detergent sulfobetaine 201 (NDSB-201; 500 mg/mL) in a 1:9 ratio. Cryo-solution B was made by combining cryo-solution A:crystallization cocktail:sucrose solution in a 10:3:7 ratio. Preliminary crystals were soaked for ~15 s in two droplets (~5 μL) composed of 25% and 50% v/v cryo-solution B diluted in crystallization cocktail, in order of increasing cryo-solution B concentration, followed by vitrification in liquid nitrogen.

The microbatch-under-oil crystallization procedure was further optimized. Crystals were grown at 14 °C and the crystallization droplet comprised 2 μL of *holo*-ObiF1 protein solution at 36 mg/ml and 1 μL of crystallization cocktail supplemented with 2 mM of β-OH-p-NO$_2$-L-homoPhe. Purified β-OH-p-NO$_2$-L-homoPhe (> 98% pure) was prepared enzymatically as described previously[18]. Briefly, p-NO$_2$-phenylpyruvate (5 mM) was treated with decarboxylase ObiG (10 μM) and transaldolase ObiH (10 μM), both from *P. fluorescens* ATCC 39502, in a buffer containing ThDP (10 μM), MgCl$_2$ (10 μM), PLP (10 μM), L-Thr (5 mM), and MES (25 mM) adjusted to a final pH of 7.5. All concentrations reflect the final concentration in a 1 mL total reaction volume. After 3 h at room temperature, the reaction was quenched by addition of TFA and pure was β-OH-p-NO$_2$-L-homoPhe was obtained after purification by preparatory HPLC (Beckman Coulter SYSTEM GOLD 127 P with 168 detector; Phenomenex Luna 10μ C18(2) 100A column; mobile phase (A) was 0.1% TFA in water; mobile phase (B) was 0.1% TFA in MeCN; gradient was 0% B held for 5 min then 0% B to 100% B over 10 min; retention time ~14 min). Crystals were cryo-protected by serial transfer through four drops (~3 μL under paraffin oil, ~30 min soak, 14 °C, in order of increasing PEG3350 concentration) of 70 mM HEPES pH 7.0, 8 mM NaCl, 1.7% v/v glycerol, and 1.7% w/v 1,3-dimethylimidazolium dimethyl phosphate with increasing PEG3350 concentration (27, 34, 40, 45% w/v). Each drop was supplemented with 2 mM β-OH-p-NO$_2$-homoPhe. Following the final soak in the 45% w/v PEG3350 solution, crystals were vitrified directly in liquid nitrogen.

**X-ray data collection and structure determination**. Diffraction data were collected at APS beamline 23-IDD (unliganded crystals) and SSRL beamline 12-2 (optimized, ligand-bound crystals). Each diffraction dataset (3.0 Å) was derived from a single crystal and exhibited moderate anisotropy. Diffraction data for the unliganded dataset was indexed, integrated, scaled, and merged in space group $P2_12_12_1$ with the XDS program package. The ligand-bound dataset was indexed and integrated in the DIALS program package followed by scaling and merging in space group $P2_12_12_1$ with AIMLESS. Due to the moderate diffraction anisotropy, processed intensities were converted to structure factor amplitudes with the program phenix.reflection_file_converter using the "massage_intensities" feature, which employs a flat prior probability distribution during the conversion of negative intensities to positive amplitudes. The ObiF1 ligand-bound model was refined against these processed amplitudes.

An initial structure was determined from the unliganded dataset through iterative molecular replacement with PHASER, as implemented in the PHENIX suite of programs. A search of the Protein Data Bank identified structures with high sequence homology to each domain of ObiF1. Specifically, the adenylation domain showed 39% sequence identity to the adenylation domain of DhbF (5U89), the condensation domain showed 28% identity to condensation domain of EntF (5T3D), and the thioesterase domain showed 29% identity to the type II thioesterase from the rifamycin biosynthetic cluster (3FLB). The domains were placed in order from the largest to the smallest domains. A search model for each domain was generated using the PHENIX Sculptor module to remove non-conserved side chains and gaps.

The adenylation domain was first placed via molecular replacement using PHASER. This was used as a fixed model in the subsequent search for the condensation domain. The two domains were positioned similar to the didomain core of prior NRPS modules. Placement of the thioesterase domain provided a convincing model and map. No effort was made to place the PCP or MbtH-like protein (MLP) domain through molecular replacement. The final molecular replacement solution of all the three domains contained 1165 residues. Removal of residues with poor density identified a core conserved 769 residues that served as a starting point for manual model building and refinement. Continued building of loops and side chains led to improved electron density, which enabled the identification of the remaining residues.

The domains of ObiF1 (Supplementary Fig. 14) are composed of residues 1–439 (C), 444–972 (A), 978–1050 (T), 1059–1303 (TE), and 1313–1370 (MLP). The conserved hinge residue that separates the N- and C-terminal subdomains of the adenylation domain is at residue Asp873. The final structural model contains a single polypeptide chain with disordered regions that occur at residues 73:76, 619:622, 924:928, 973:977, 1051:1058, and 1304:1312. These regions correspond primarily to domain-spanning residues (A–T domains, 973:977; T–TE domains,

1051:1058; TE–MLP domains, 1304:1312), or loop residues within the C domain (73:76) or the A domain C-terminal subdomain, also called the $A_{sub}$ domain (924:928). Disorder within these regions is consistent with the dynamic domain configuration that is required to enable the PCP domain to visit the neighboring catalytic domains within the module.

Protein atoms from the unliganded ObiF1 protein were used as a molecular replacement search model to determine the structure of the protein bound to the β-OH-*p*-NO₂-L-homoPhe substrate. Three separate search models were generated from individual domains of the unliganded ObiF1 structure. Domains were placed in order of A+MLP domain (residues: 460–872:1329–1371), C domain (residues 5–444), and TE domain (residues: 1059–1303). The PCP domain and resolvable linker regions were then built in manually with iterative real space refinement in Coot. Restraints for β-OH-*p*-NO₂-L-homoPhe and phosphopantetheine were generated with eLBOW[39]. The liganded ObiF1 structure was further refined with PHENIX refine. Final rounds of refinement employed translation-libration-screw (TLS) parameterization; 9 TLS domains defined at residues 4:169, 170:379, 380:453, 454:616, 617:776, 777:870, 1059:1131, 1132:1303, and 1313:1370 were determined with the TLSMD server[40]. Initially, 12 TLS domains were determined with the TLSMD server; three of these initial domains corresponding to the A domain C-terminal subdomain and thiolation domain were excluded from TLS refinement. Final refined TLS matrices were confirmed to correspond to harmonic motion of rigid bodies with the program PHENIX tls_analysis[41]. Diffraction and refinement statistics are displayed in Supplementary Table 2.

**Mutagenesis of ObiF1.** PCR-based mutagenesis was used to produce *obiF1* mutants encoding A841E, C1146A, C1146S, G1173L, D1177A, R1190A, D1254A, R1263A, H1284A, D1254A/H1284A, and G1173D/D1254A proteins. In addition to the A841E mutant, we designed two additional constructs to validate the interaction of the ObiF1 MLP and adenylation domains. A truncated del1304 protein that lacked the entire MLP domain was produce by mutagenesis to introduce a stop codon after the codon for Leu1303. A second construct designated rbs-1303/1304 introduced a stop codon, a ribosome binding site, and an ATG codon between the TE and MLP domains to co-express the ObiF1 C-A-PCP-TE and MLP proteins. Finally, we designed the ObiF1-MLP-1315 construct to produce the ObiF1 MLP protein consisting of residues Ala1315–Val1390 for supplementation assays. Following the Val1390 residue, this construct encoded a purification tag with the amino acid sequence of KLAAALEHHHHHH.

Mutagenesis primers were produced by Integrated DNA Technologies (Coralville, IA, USA; Supplementary Table 3). Phusion High-Fidelity DNA Polymerase (ThermoFisher) was employed for PCR amplification with a two-step amplification protocol (1 × 3 min 98 °C; 30 × 0.5 min 98 °C, 11.5 min 72 °C). PCR reaction mixtures (50 µL total volume) were composed of DMSO (4% v/v), forward/reverse mutagenesis primers (200 nM each), Phusion GC (C1146A, C1146S, G1173L, D1177A, R1190A, D1254A, R1263A, H1284A, D1254A/H1284A, G1173D/D1254A, del1304) or HF (A841E, rbs-1303/4) buffer (1×), dNTPs (1 mM total), pET28-ObiF1 template (10 ng), and Phusion DNA polymerase (1 U). Polymerase incomplete primer extension (PIPE)[42] was employed for mutants rbs-1303/4 and del1304. For these two mutants, separate PCR reaction mixtures (50 µL) were setup for both forward and reverse primers (400 nM each). The reactions were run for one cycle, combined (1:1; 50 µL total volume), and then run for an additional 29 cycles.

Following completion of PCR amplification, DpnI (0.5 µL) was added to reaction mixtures, which were incubated at 37 °C for ~1 h. PCR products were purified by gel electrophoresis (0.9–1% w/v agarose) and then transformed into NEB Turbo competent *E. coli* cells (high efficiency). Clones were selected for on LB agar containing kanamycin (50 µg/mL). For double mutants D1254A/H1284A and G1173D/D1254A, mutagenesis PCR and transformation procedures were performed iteratively using pET28-ObiF1-D1254A as a template for the second reaction. ObiF1-MLP-1315 was PCR amplified from a pET28-ObiF1 template and cloned into a pET24 vector (NdeI/HindIII) with a C-terminal 6xHis-tag. Plasmids were then isolated and sequenced to establish the presence of the desired mutation(s). Sequence-confirmed plasmids were then used to transform competent BL21(DE3) *E. coli* cells for protein expression. The ObiF1-WT and TE domain mutants (C1146A, C1146S, G1173L, D1177A, R1190A, D1254A, R1263A, H1284A, D1254A/H1284A, G1173D/D1254A) were produced in BL21(DE3) for purification. In contrast, the wild-type ObiF1, the ObiF1 MLP protein, and mutant ObiF1 enzymes used to test the role of the MLP interaction (A841E, rbs1303/4, del1304) were expressed in BL21(DE3)ΔybdZ (courtesy of Prof. Michael G. Thomas, University of Wisconsin), which lacks the only endogenous *E. coli* MLP gene.

**Purification of ObiF1 variants for enzyme reactions.** The wild-type ObiF1 and TE domain mutants (C1146A, C1146S, G1173L, D1177A, R1190A, D1254A, R1263A, H1284A, D1254A/H1284A, G1173D/D1254A) were purified using a slightly different protocol than the ObiF1 used for structural studies. One liter of TB media (30 µg/mL kanamycin) was inoculated with 1 mL of an LB (30 µg/mL kanamycin) overnight culture (37 °C, 250 rpm) of *E. coli* BL21(DE3) cells harboring WT or mutant plasmid. The TB culture was grown (37 °C, 250 rpm) until the culture reached an $OD_{600}$ of ~0.6 at which time IPTG (500 µM) was added and the culture was incubated for 20 h (15 °C, 250 rpm). Cells were harvested via

centrifugation (5000 × *g*, 35 min, 4 °C) and flash frozen in liquid nitrogen. Subsequent purification steps were performed at 4 °C. Frozen cell pellets were thawed and re-suspended in lysis buffer (50 mM KH₂PO₄ pH 8.0, 500 mM NaCl, 0.2 mM TCEP, 20 mM imidazole, 10%v/v glycerol; 20 mL lysis buffer per gram cell pellet). Cells were lysed by mechanical disruption (Branson Sonifier) and the resulting cell lysate was clarified via centrifugation (40,000 rpm, 40 min, 45-Ti rotor), filtered (0.45 µm), and passaged over a Ni-loaded HisTrap (5 mL) column. The column was washed with 5 column volumes of 10% v/v elution buffer (50 mM KH₂PO₄ pH 8.0, 500 mM NaCl, 0.2 mM TCEP, 300 mM imidazole, 10%v/v glycerol) in lysis buffer and immobilized protein was eluted with 6 column volumes of elution buffer. Fractions containing ObiF1 were combined and pantetheinylated as described (*vide supra*). Pantetheinylation reactions were then purified via gel filtration (GE Healthcare HiLoad 16/600 Superdex 200, 0.8 mL/min). Gel filtration of ObiF1-WT was carried out in 50 mM KH₂PO₄ pH 8.0, 150 mM NaCl, and 0.2 mM TCEP. Gel filtration of ObiF1 TE domain mutants was carried out in 50 mM KH₂PO₄ pH 8.0, 500 mM NaCl, 0.2 mM TCEP, and 10% v/v glycerol. Following gel filtration, fractions containing ObiF1 mutants were pooled (6–9 mL total volume) and dialyzed (12–14 kDa MWCO) for 20 h against 1 L of 50 mM KH₂PO₄ pH 8.0, 150 mM NaCl, and 0.2 mM TCEP. Following gel filtration (ObiF1-WT) or dialysis (ObiF1 TE domain mutants), samples were concentrated to 21–28 mg/mL (15 mL Amicon Centrifugation filter, 50 kDa MWCO; G1173L mutant was concentrated to 8 mg/ml), flash frozen in liquid nitrogen, and stored at –80 °C.

**Protein purification for MLP studies.** Wild-type ObiF1 and the ObiF1 MLP domain mutants (A841E, rbs1303/4, del1304), as well as the free-standing MLP (ObiF1-MLP-1315) were purified for in vitro enzyme reactions. One liter of TB media (kanamycin; 30 µg/mL ObiF1-WT, ObiF1 mutants; 50 µg/mL ObiF1-MLP-1315) was inoculated with 1 mL of an LB (kanamycin; 30 µg/mL ObiF1-WT, MLP domain mutants; 50 µg/mL MLP-1315) overnight culture (37 °C, 250 rpm) of *E. coli* BL21(DE3)ΔybdZ cells harboring WT or mutant plasmid. Subsequent expression and purification steps were performed as described for ObiF1 TE domain mutants (cell pellets harboring ObiF1-MLP-1315 were re-suspended in 10 mL lysis buffer per gram cell pellet). Following gel filtration, fractions containing the desired protein were pooled, concentrated directly in gel filtration buffer (21–27 mg/mL ObiF1-WT and MLP domain mutants, 15 mL Amicon Centrifugation filter, 50 kDa MWCO; 6 mg/mL MLP, 4 mL Amicon Centrifugation filter, 10 kDa MWCO), flash frozen in liquid nitrogen, and stored at –80 °C.

**Purification of additional enzymes for biochemical assays.** Genes encoding ObiF2 (WP_059467197.1), ObiH (WP_059467200.1), and ObiD (WP_059467195.1), were synthesized, codon optimized (Supplementary Figs. 15–17), and cloned into a pET28 vector (NdeI/HindIII) encoding for an *N*-terminal hexahistidine tag (Genscript; Piscataway, NJ). ObiF2, ObiD, and ObiH were expressed with *N*-terminal hexahistidine tags from 1 L cultures of *E. coli* BL21 (DE3) in TB medium supplemented with 50 µg/mL kanamycin. Expression cultures were inoculated from a 5 mL overnight culture of *E. coli* BL21 (DE3) harboring the appropriate plasmid and grown to OD600 ~0.4. Cultures were then cooled in an ice bath prior to addition of a sterile IPTG solution (0.1 mM IPTG final concentration). Cultures were then incubated at 15 °C with agitation for ~18 h. Cells were cooled to 4 °C (all following steps were performed at 4 °C) and harvested via centrifugation at 5000 rpm for 20 min. The supernatant was discarded and cell pellets were suspended in 40 mL of lysis buffer (50 mM K₂HPO₄, 500 mM NaCl, 5 mM β-mercaptoethanol, 20 mM imidazole, 10% glycerol, pH 8.0) and flash frozen in liquid nitrogen. After thawing, cells were lysed by two passages through an Avestin EmulsiFlex-C5 cell disruptor. Cell lysate was clarified via centrifugation at 45,000 rpm for 35 min. Supernatant was treated with Ni-NTA resin for 30 min. After two washings with lysis buffer the proteins were eluted with 5 × 10 mL aliquots of elution buffer (50 mM K₂HPO₄, 500 mM NaCl, 5 mM β-mercaptoethanol, 300 mM imidazole, 10% glycerol, pH 8.0). Elution fractions were analyzed by SDS-PAGE with Coomassie blue visualization. Fractions containing pure proteins were combined and dialyzed into phosphate buffer (50 mM K₂HPO₄, 150 mM NaCl, 1 mM DTT, pH 8.0) using 10,000 MWCO SnakeSkin dialysis tubing. Protein solutions were concentrated using Amicon 10,000 MWCO centrifugal filters followed by flash freezing as ~50 µL beads and storage at –80 °C.

**In vitro ObiF1/F2/D/H enzyme reactions.** For ObiF1 mutant assays the thiol trapping agent was GSH, the aldehyde substrate was PAA (**1a**), and the benzoic acid substrate was 2,3-DHB (**3a**). For all ObiF1 substrate screens the thiol trapping agent was DTT. For aldehyde (**1a–h**) substrate screens the benzoic acid substrate was 2,3-DHB (**3a**). For benzoic acid substrate screens (**3a–u**) the aldehyde was PAA (**1a**). All product nomenclature and *m/z* values assigned for each product are provided in Fig. 6 and Supplementary Data 1 captions. To ensure complete pantetheinylation, ObiF1 variants and apo-ObiD were pantetheinylated at room temperature for 2 h by combining 180 µM CoASH, 5 mM thiol, 10 mM MgCl₂, 1 µM Sfp, 25 µM *apo*-ObiD or ObiF1 variant, and 25 mM sodium phosphate buffer, pH 7.5. For all reactions, ObiH at 5 µM was first incubated at room temperature with 500 µM of the aldehyde substrate along with 1 mM L-Thr, 500 µM thiol, and 5 µM PLP in 25 mM sodium phosphate buffer, pH 7.5 (final working concentrations). After 1 h, 1 µM *holo*-ObiF1, 1 µM ObiF2, 1 µM *holo*-ObiD, 5 mM ATP, and

500 μM benzoic acid (final working concentrations) were added to the ObiH reaction giving a final working volume of 300 μL for each experiment. For WT, del1304, rbs1303/4, and A841E additional experiments were performed with added MLP domain at a final concentration of 1, 10, or 25 μM. Reactions were allowed to proceed at room temperature and 100 μL aliquots for LCMS analysis were quenched at 30 min, 3 h, and 24 h time points by addition of 1 M HCl to give a final pH ~2. LCMS samples were spiked with 100 μM phenylalanine (final concentration) as an internal standard and frozen at −80 °C. At the time of LC-MS analysis, the sample was thawed, centrifuged at 16,000 r.p.m. for 1 min to pellet solids, and analyzed by LCMS (instrument: Agilent 6130 quadrupole with G1313 autosampler, G1315 diode array detector, 122 series solvent module; column: Phenomenex Gemini C18, 50 × 2 mm, 5 μm plus guard column; solvents: 0.1% formic acid in (A) water and (B) acetonitrile; method: 5% B to 100% B over 20 min; software: G2710 ChemStation). All experiments with ObiF1 mutant comparison and MLP supplementation were performed in at least duplicate as independent trials. All experiments with ObiF1 substrate screens were performed as a single trial. Extracted ion counts for each product were plotted either as bar graphs (ObiF1 mutant studies; error bars represent standard deviations) or heat maps (ObiF1 substrate screens) using GraphPad Prism v7. Product assignments were confirmed by high-resolution mass spectrometry collected using an LTW-Velos Pro Orbitrap at the Donald Danforth Plant Science Center, St. Louis, MO. Note: PAA used for WT ObiF1 and D1254A, H1284A, D1254A/H1284A, C1146S, C1146A, D1177A, R1263A, R1190A, G1173D/D1254A, and G1173L was purchased from MilliporeSigma (St. Louis, MO). During the course of revisions, PAA went on back order from the supplier. Therefore, PAA used for the WT ObiF1 and del1304, rbs1303/4, and A841E assays was synthesized from phenethyl alcohol using Dess-Martin periodinane oxidation in anhydrous $CH_2Cl_2$ followed by purification via silica gel column chromatography (5% EtOAc, 95% Hex) to give PAA in > 95% purity with NMR characterization data matching authentic material from MilliporeSigma.

**Reporting summary**. Further information on research design is available in the Nature Research Reporting Summary linked to this article.

## Data availability
The coordinates and structure factors for BdObiF1 are available from the Protein Data Bank, accession 6N8E. The LCMS chromatograms underlying Figs. 4c, 4d, 6b, and 6c are provided in Supplementary Data 1. The source data underlying Supplementary Fig. 8 are provided as a Source Data file. Other data are available from the corresponding authors upon reasonable request.

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

## Acknowledgements

T.A.W. would like to acknowledge support for this research from the Research Corporation for Science Advancement through a Cottrell Scholar award and the Alfred P. Sloan Foundation through a Sloan Fellowship. We thank B. Evans at the Proteomics & Mass Spectrometry Facility at the Donald Danforth Plant Science Center (St. Louis, MO) for HRMS acquisition (NSF Grant No. DBI-0521250). A.M.G. acknowledges the support of the National Institutes of Health (NIGMS Grant GM116957). Diffraction data were collected at the GM/CA ID-B beamline of Advanced Photon Source, Argonne National Lab. GM/CA@APS is funded in whole or in part with Federal funds from the National Cancer Institute (ACB-12002) and the National Institute of General Medical Sciences (AGM-12006). This research used resources of the Advanced Photon Source, a U.S. Department of Energy (DOE) Office of Science User Facility operated for the DOE Office of Science by Argonne National Laboratory under Contract No. DE-AC02–06CH11357. The Eiger 16M detector was funded by an NIH–Office of Research Infrastructure Programs, High-End Instrumentation Grant (1S10OD012289-01A1).

## Author contributions

A.M.G. supervised the structural experiments. T.A.W. supervised the functional experiments. D.F.K. designed, purified, and crystallized ObiF1, and solved and refined the BD-ObiF protein structure. D.F.K. cloned, expressed, and purified ObiF1 mutants. J.E.S. and E.M.G. performed all functional studies of ObiF1 mutants and provided purified β-hydroxy-p-nitro-homoPhe for co-crystallization studies. J.E.S. and E.M.G. expressed and purified ObiH, ObiD, and ObiF2. D.F.K. cloned, expressed, and purified the ObiF1 MLP domain. E.M.G. performed all functional studies of ObiF1 substrate screens and MLP supplementation. The structure and functional results were analyzed by D.F.K., J.E.S., E.M.G., T.A.W., and A.M.G. The supplementary information document was prepared by D.F.K., J.E.S., E.M.G., T.A.W., and A.M.G. The Supplementary Data 1 was prepared by E.M.G., J.E.S., and T.A.W. The paper was written by D.F.K., T.A.W., and A.M.G with input on experimental methods and figures from J.E.S. and E.M.G.

## Additional information

**Competing interests:** The authors declare no competing interests.

