## [Peer Review File · Nature Communications]

Reviewers' comments:

Reviewer #1 (Remarks to the Author):

I found this to be an interesting paper that includes the first structural characterisation of a modular NRPS with several interesting features, including a fused MbtH like protein that displays an apparently novel interaction with the adenylation domain; a large A-domain substrate that can be seen within the AA binding pocket; and a TE domain that catalyses the cyclisation of the b-OH group to generate a 4-membered lactone on cleavage. An investigation of the mechanism of the TE domain and an analysis of the substrate scope of the various catalytic domains are also included in the work.

Main sections of the paper can be broken down into:

1. Structural characterisation. In general I found the structural work to be convincing, although the discussion of the structure, comparison to other structures and figures could do with improvement as currently the structure is not well integrated with the literature. Structural and sequence alignments would assist, along with careful choice of structural views, indicating distances and larger sizes. Density of the A-domain bound ligand and representation of the pocket requires at least one further (top) view, and the MLP interface would also benefit from comparisons to other MLP-bound A-domain structures. Another aspect that is mentioned in the abstract but not really addressed in the main text is the dynamics of the NRPS machinery, which could be achieved through discussion of the current modular structure with reference to the others now known.

2. Role of the MLP. The investigation of the role of the MLP could be improved in terms of the necessity to retain the MLP in a fused form. Experiments in which the MLP was coexpressed as a separate protein would be most informative as to the role of the fusion, and these experiments would also help to further support the experimental data around the Ala->Glu mutant discussed. I would also think alternative mutations could also provide a more comprehensive overview of the role of this MLP. Supplementation with a related MLP sequence in trans would also be a most interesting follow up experiment.

3. Mechanism of the TE domain. A significant number of mutants have been made and characterised: here, my concern is that the use of the thiol capture assay to infer production of the 4-membered lactone ring is not able to resolve any effects that these mutants may have had on the mechanism of the TE domain. For example, this could involve thioester exchange or simple hydrolysis of the precursor from the TE domain. I would suggest that direct measurement of the lactone product using short reaction times could help to allay these concerns. A more mechanistic figure could help as well, along with more comparisons to appropriate literature including the work of Townsend on nocardicin (both TE and C-domain activity) and Sherman (PKS TE Cys/Ser mutations).

4. Promiscuity of the biosynthetic system. Here, there is a lot of work presented that involves the characterisation of the selectivities of the trans-aldolase enzyme that makes the b-OH AA precursor as well as all NRPS domains (2x adenylation domains, condensation domain and TE domain). The problem with interpreting the results is that there may be different specificities of the different domains, and thus it is difficult to really assess the data. Taking some of the enzyme reactions out of "one pot" could assist here (ObiH, ObiF2 loading of T_ar). I also feel that the results do not indicate a high degree of promiscuity in substrate selection, but rather that they support some ability to tune the substrate structure around the parent compound.

Overall, I felt that the discussion was rather brief and did not do justice to the extensive experimental work within the results section. A more complete discussion would also be most helpful for the non-expert reader to be able to better understand the implications and significance of these results.

Specific points to consider together with those mentioned above include:

Figure 1 - should have domain arrangement indicated above to show how the domains are arranged from N-C termini (include like in 2)

Lack of information about NRPS cycling and reactions - more explanation help the non-expert? Mentions "four different stages of the NRPS catalytic cycle" but this is not displayed/ illustrated currently

T/PCP - indicate possibility for both naming nomenclatures

Figure 2 - panels C and D are far too small - no distances indicated in the figure

Lack of comparison of this structure to the 5 previous ones indicated. This goes along with the lack of discussions around domain dynamics

Lack of discussions round MLP vs other structures; also some indications in the text that the MLP interaction with the A-domain is the same as previously, which seems to contradict other parts that indicate this is different...?

"Specificity-conferring sequences comprise residues that form the hydrophobic pocket in which the substrate binds and are thus indifferent towards residues that form specific interactions with polar or charged functional groups along the substrate backbone." I do not think this is always true - this indicates (or to me at least implies) that no residue in the pocket is involved in H-bonding or salt bridging interactions with the substrate at all. See example in Kaljunen et al ACIE 2015.

Figure 3b - should show full pocket and interactions; need better views of the substrate as well. "The side chain of Trp666, which adopts two alternate conformations, contributes to the hydrophobic pocket around the p-NO₂-phenyl ring" - show this. Distances should also be shown. Out of interest, what steps have been taken to "verify" the ligand density? Show difference density or a Polder map?

"Perhaps the BD-ObiF1 MLP promotes formation of the adenylate forming conformation of the A domain" - speculation; either support or remove. In vitro assays with/ without MLP in trans could assist here.

MLP comparison of sequence should be done in a figure.

"Therefore, there may be a subtle difference in MLP-A domain interactions depending upon whether the MLP domain is upstream or downstream of the A domain." Same comment as above - support with evidence or remove the speculation. Trans interaction assays could help.

MLP interactions in figure 3 c/d very hard to understand; figure needs to be reconsidered.

"This reduction in protein yield suggests that the MLP interaction with the ObiF1 adenylation domain does influence protein stability." Could also simply be a folding issue before the MLP is even translated. Can't say without more evidence that it is the loss of the MLP interaction that causes the loss in activity; it could be protein misfiling. Other mutants should be tested that are less extreme than Ala to Glu along with the idea to remove the MLP and include it in trans.

Why not reconstitute the whole system for the TE activity assays? Control cleavage of loaded starting material to establish conversion? Why the 24 hr timepoint?

Conversion of Cys to Ser? reference to Sherman's work on TE characterisation during PKS biosynthesis again with a Cys-containing TE? Compare structure also to Schmeing/Chin's new trapped TE structure in Nature. Discussion of nocardicin TE domain would also seem appropriate.

Figure 5 - details of superposition / RMSD etc. are needed

R1190 - could it also be involved in PCP binding to the TE?

Figure 6 is very confusing and the data seems to be less indicative of general promiscuity but rather some scope for modification around the normal substrates.

Why is DTT used for the substrate acceptance experiments and not GSSG? What are the differences if both are used on the same assay?

" In most cases, the β -hydroxy- α -amino acids (2a-h) can be detected by LCMS (Figure 6b) indicating that ObiH is broad in substrate scope." I do not agree with this - the range of substrates used is small and many do not seem to work. Why not run this assay on its own as well to see what accumulates? Very hard to work out the selectivities of one enzyme amongst the activities of several enzymes. One option for some of the amino acid substrates would be to solve a co-crystal structure with these bound in the A-domain pocket to help justify some of the speculations made here. One could also make the A-domain assays competitive to see if the domain favours one substrate over the other.

Discussion:

There seems to me to be quite a bit of speculation in here; plus a narrative that links the work is somewhat lacking. Discussion of different substrate preferences - has this been tested? Nothing in here about the TE from nocardicin; nothing about other TE domains from NRPS systems that have been shown to act against different peptides. I do not think this TE domain is really a logic gate (or at least as I understand it) - a logic gate TE is the case when there are late stage modifications required from other enzymes prior to the TE becoming active; here the TE displays a different cleavage mechanism, but is not a logic gate. There is also no discussion of C-domain mediated β -lactam formation, which I would have expected.

Reviewer #2 (Remarks to the Author):

The paper by Kreitler et al describes the structure of the obafluorin forming NRPS ObiF1 that carries a MLP-domain as C-terminal domain. Overall, the unique biosynthesis of NRPS-derived β -lactones and the widespread occurrence of such NRPS enzymes (see Scott et al) clearly justifies the paper, which is well written by experts in the field of NRPS structural biology. I really enjoyed reading the manuscript that contains a lot of interesting information regarding all the involved NRPS domains and gives a lot of new insights into this unusual type of NRPS biochemistry and NRPS in general.

However, I have a few comments that should be addressed by the authors:

- Please correct and check *P. fluorescens* in the manuscript
- I would suggest to add representations of both subtypes of ObiF enzymes and the two BGCs in both known producers to the SI
- Page numbers are missing which make the paper difficult to review. But Bode was not an author in the paper describing the characterization of SlgN1 mentioned on p. 14? Either the citation or the authors must be corrected (the author I guess)
- P.13: MLP consensus sequence: The similarity/differences are hard to see in the current form and I would suggest an alignment in the SI showing several sequences
- A841E mutation shows a loss of ObiF1 activity but have the authors tried to cut the MLP domain off (and complement with it in trans?) This would be especially interesting with respect to the claims made in the discussion about the A domain specificity.
- Can stand-alone MLPs substitute the cis MLP function?
- Along this line: what about the A domain-only specificity? Fig 6 shows the activity of a full ObiF1 protein, which might be different compared to the A domain.
- For me, it is clear that the domain ObiF1 domain organization is unique but It might be possible to get some knowledge especially about the MLP function with some minor experiments as suggested above.
- Fig. 4: C and D are not good to read. Would a log scale be better? What is the detection limit of both compounds in this MS system?
- Fig. 6. The heatmap is a good representation of such a number of experiments that I like a lot. However, the ionization of the different compounds is not considered in such a representation and there might be huge differences especially for amine or halogen containing derivatives. Since the

wt derivatives or closest relatives in both cases are produced best, it is in principle not necessary to compare the ionization but I would be careful using only signal intensity without making standard curves for the different compound classes or checking the ionization of the used acids. The amount of some derivatives might in fact be even much lower but suggesting a higher amount from a much better ionization (for example for af).

- Were other than benzoic acids tested as substrates in this assay? Is the A domain specificity of ObiF2 known? What about cinnamic acid or non-aromatic acids?

- Supplementary note:

o Make all figures and tables understandable without the introduction of p.2 so please add all explanations to the figure/table captions (like NA etc)

- Supporting information:

o Supp Fig 2: why is there such a large peak of 2a already at 0 min

o Supp Fig 2-4: I would suggest to use a slightly different color code not changing the color of the carbonyl group but for example showing H₂O in complete blue (comp. Supp Fig 4)

o Supp Fig 5: some derivatives seem to fluctuate in time like fa or ha. Have these experiments been done multiple times or are these just minor differences?

- final general question: What can be said about the interaction between ObiD and ObiF that is essential for product formation? Are there COM or docking domains (especially at the N-terminus of ObiF1)? Is there a stable complex formed between ObiD and ObiF1F2? Has a pull down experiment been performed that might point to a stable interaction?

Reviewer #3 (Remarks to the Author):

Gulick, Wenczewicz and coworkers present structural and biochemical investigation of the obafluorin synthetase system. *Burkholderia diffusa* obafluorin synthetase subunit F1 has an unusual domain organization of C-A-T-TE-MLP, and has a TE domain that catalyzes a release step by cyclization to a β -lactone. The authors determined a crystal structure of *B. diffusa* ObiF1 and performed mutagenic and specificity analyses using a reconstituted obafluorin synthesis assay.

Overall, this is a very nice study that will add to the growing structural understanding of NRPS biology. The mutagenesis of the TE domain is nice, but without a substrate bound, it is difficult to fully understand why ObiF1 TE performs lactonization.

The most surprising finding is that in the structure of ObiF1, the MLP domain is bound to A1, implying that when all proteins are present, the MLP domain will bind A1 and not AAr of ObiF2. This A1-MLP interaction is said to anchor the TE domain to the core of the module, which may be advantageous, but as is pointed out with AB3403, SrfAC and EntF examples, does not appear to be generally necessary. The authors test the importance of this interaction by an A841E mutation which should disrupt the ObiF1 A:MLP interaction, and which is shown to markedly decrease activity. However, this experiment cannot rule out that the A841E mutation is simply so destabilizing that the A domain is partially misfolded and therefore inactive only because of misfolding, not because of lack of MLP binding. It would be very important and powerful to add some simple experiments with an ObiF1 C-A-T-TE construct, an ObiF1 MLP construct, and ObiF2 AAr: Does extracted ObiF1 MLP bind ObiF1 C-A-T-TE and/or ObiF2 AAr? Does ObiF1 MLP increase the adenylation activity of ObiF1 C-A-T-TE and/or ObiF2 AAr? (It would also be informative to know if ObiF2 bound ObiF1 in presence or absence of ObiD).

The result that D1254A generated approximately 25% as much β -lactone as wild type seems higher than expected. Can the authors comment on this and compare to available mutational data for other TE domains?

I suggest shortening the "EntF-like triad configuration inhibits ObiF TE1 domain substrate loading" section. It is not surprising that simple reintroduction of ancestral residues into an evolutionarily diverged active site would not be sufficient to impart ancestral / full activity. Likewise, it is not uncommon that two mutations (G1173D/D1254A) are more deleterious than one (G1173D).

Minor:

Abstract: "Nonribosomal Peptide Synthetase (NRPS)"

-The term nonribosomal peptide synthetase is not a proper noun and shouldn't be capitalized (It is not capitalized in the introduction.)

"The NRPS antimicrobial β -lactone obafluorin is produced from a nonproteinogenic β -hydroxy amino acid"

-Reword to clarify that the β -hydroxy amino acid is not the sole substrate as DHB is also a substrate.

Introduction: "...differ significantly from known BGCs."

-Perhaps rephrase to "from well-characterized BGCs."

"ObiF1/F2 module"

-Is ObiF1/F2 a module?

Results, page 11: "L-Thr is the simplest analog that mimics the β -OH-p-NO₂-homoPhe backbone stereochemistry and is not accepted as a substrate by the ObiF1 A domain. Therefore, structural features shared between Thr1 and the ObiF A domain are likely to be common among β -hydroxy acid adenylating enzymes."

-I'm unsure why the inability of the ObiF1 A domain to accept L-Thr is indicative of common features.

Results: "In line with this hypothesis we observed a hydrogen bonding network of the catalytic triad at residues Cys1146, His1284, and Asp1254 (Figure 4b)."

-Please show the hydrogen bonds in Figure 4b.

Reviewers' comments:

Reviewer #1 (Remarks to the Author):

I found this to be an interesting paper that includes the first structural characterisation of a modular NRPS with several interesting features, including a fused MbtH like protein that displays an apparently novel interaction with the adenylation domain; a large A-domain substrate that can be seen within the AA binding pocket; and a TE domain that catalyses the cyclisation of the b-OH group to generate a 4-membered lactone on cleavage. An investigation of the mechanism of the TE domain and an analysis of the substrate scope of the various catalytic domains are also included in the work.

Main sections of the paper can be broken down into:

1. Structural characterisation. In general I found the structural work to be convincing, although the discussion of the structure, comparison to other structures and figures could do with improvement as currently the structure is not well integrated with the literature. Structural and sequence

alignments would assist, along with careful choice of structural views, indicating distances and larger sizes. Density of the A-domain bound ligand and representation of the pocket requires at least one further (top) view, and the MLP interface would also benefit from comparisons to other MLP-bound A-domain structures. Another aspect that is mentioned in the abstract but not really addressed in the main text is the dynamics of the NRPS machinery, which could be achieved through discussion of the current modular structure with reference to the others now known.

We have added to the text to provide more structural details as requested by the reviewer. Please also see responses to points 6, 9, 12, and 14 below

2. Role of the MLP. The investigation of the role of the MLP could be improved in terms of the necessity to retain the MLP in a fused form. Experiments in which the MLP was coexpressed as a separate protein would be most informative as to the role of the fusion, and these experiments would also help to further support the experimental data around the Ala->Glu mutant discussed. I would also think alternative mutations could also provide a more comprehensive overview of the role of this MLP. Supplementation with a related MLP sequence in trans would also be a most interesting follow up experiment.

All three reviewers have raised this concern in one form or another. We have performed an elaborate series of experiments that confirm the interaction of the MLP with the upstream adenylation domain. The results of these experiments are described in detail in the text (Figure 4B and pg 14-17). Please also see response to point 17 below.

3. Mechanism of the TE domain. A significant number of mutants have been made and characterised: here, my concern is that the use of the thiol capture assay to infer production of the 4-membered lactone ring is not able to resolve any effects that these mutants may have had on the mechanism of the TE domain. For example, this could involve thioester exchange or simple hydrolysis of the precursor from the TE domain. I would suggest that direct measurement of the lactone product using short reaction times could help to allay these concerns. A more mechanistic figure could help as well, along with more comparisons to appropriate literature including the work of Townsend on nocardicin (both TE and C-domain activity) and Sherman (PKS TE Cys/Ser mutations).

We have monitored the functional assays for β -lactone formation using a thiol capture that addresses two problems. First, the β -lactone does not ionize well under electrospray ionization in positive ion mode, which introduces a larger degree of error. Second, the β -lactone hydrolyzes quickly ($t_{1/2}$ ~5min). These problems are solved by the thiol trapping agent, which has a greater ionization potential and much greater stability towards hydrolysis ($t_{1/2}$ ~3hr). We have validated the thiol trapping agent as a reliable proxy for β -lactone using purified obafluorin and the hydrolysis product from *Pseudomonas fluorescens* as described in the revised text and supplementary information (Figures S5-7). Only the β -lactone reacts instantaneously and quantitatively with a thiol trapping agent. Furthermore, the C1146A mutant represents a baseline for direct transthioesterification cleavage from the pPant of the NRPS to the exogenous thiol, which is much slower than β -lactone trapping by a thiol as demonstrated by the control reactions with pure obafluorin. The thiol capture assay is the superior method for comparing the NRPS mutants. We have revised Figure 4 to better highlight the NRPS release mechanisms along with improved discussion in the main text. Please also see response to point 19 below.

4. Promiscuity of the biosynthetic system. Here, there is a lot of work presented that involves the characterisation of the selectivities of the trans-aldolase enzyme that makes the b-OH AA precursor as well as all NRPS domains (2x adenylation domains, condensation domain and TE

domain). The problem with interpreting the results is that there may be different specificities of the different domains, and thus it is difficult to really assess the data. Taking some of the enzyme reactions out of "one pot" could assist here (ObiH, ObiF2 loading of T_ar). I also feel that the results do not indicate a high degree of promiscuity in substrate selection, but rather that they support some ability to tune the substrate structure around the parent compound.

We have performed stand alone ObiF reconstitution assays using the enzyme from *Pseudomonas fluorescens* and purified β -hydroxy-*p*-nitro-homoPhe in our previous paper (Schaffer, J.; et al. NCB 2016). We have validated that the coupled ObiF1/F2/D/H assay gives the same results as studying ObiF alone. The advantage of this coupled enzyme assay is the use of more readily available starting materials, which allowed for a much more comprehensive analysis. In our hands, there is no difference in the coupled enzymes assays vs stand alone enzyme assays. We have looked at acyl adenylate levels for the benzoic acids as a way to distinguish between ObiF1 and ObiF2 adenylation activity as described in the revised main text and supplementary information. Please also see responses to points 22-25 below

Overall, I felt that the discussion was rather brief and did not do justice to the extensive experimental work within the results section. A more complete discussion would also be most helpful for the non-expert reader to be able to better understand the implications and significance of these results. Specific points to consider together with those mentioned above include:

Thank you. The Discussion is now reorganized to better describe the novel MLP interaction, the unusual triad organization, and the ability of the TE domain to catalyze β -lactone formation.

5. Figure 1 - should have domain arrangement indicated above to show how the domains are arranged from N-C termini (include like in 2)

In the interest of space, we have left the linear representation of the ObiF1 protein in Figure 2. We feel that the globular domain representation reflects the advances of the structural characterization of NRPS enzymes and that it is important to begin to reflect this in the graphical depiction of these proteins. However, we have added Supplementary Figure 1, which illustrates the operon and NRPS organization for *B. diffusa* and *P. fluorescens*.

6. Lack of information about NRPS cycling and reactions - more explanation help the non-expert? Mentions "four different stages of the NRPS catalytic cycle" but this is not displayed/ illustrated currently

We have added a paragraph to the introduction (Pg. 3-4) that describes the NRPS structural cycle.

7. T/PCP - indicate possibility for both naming nomenclatures

Thank you. We have used PCP throughout the text of the manuscript and used "T" in several figures for better placement where space was constraining.

8. Figure 2 - panels C and D are far too small - no distances indicated in the figure

We have re-organized the figure to a two-column width, which allowed us to increase the size of panels C and D. We did not include distances, which we believe clutters the figure.

9. Lack of comparison of this structure to the 5 previous ones indicated. This goes along with the lack of discussions around domain dynamics

We include now Supplementary Table 1 that lists structural comparisons of ObiF1 with numerous prior structures including the full module NRPS enzymes. Because the relative orientations of the domains differs between the NRPS modules, we have chosen to present the RMS displacement values of individual domains. Upon describing the structure at the beginning of the Results section, we note the existing structures that have the highest similarity.

10. Lack of discussions round MLP vs other structures; also some indications in the text that the MLP interaction with the A-domain is the same as previously, which seems to contradict other parts that indicate this is different...?

The core interactions, namely the Trp pocket formed by the MLP that surround the similarly conserved alanine residue of the adenylation domain, are conserved. The interactions at the termini of the MLP differ. We have revised the manuscript to make that point more clearly on the bottom of pg 12.

11. “Specificity-conferring sequences comprise residues that form the hydrophobic pocket in which the substrate binds and are thus indifferent towards residues that form specific interactions with polar or charged functional groups along the substrate backbone.” I do not think this is always true - this indicates (or to me at least implies) that no residue in the pocket is involved in H-bonding or salt bridging interactions with the substrate at all. See example in Kaljunen et al ACIE 2015.

Here we were speaking specifically about the “Stachelhaus code” residues of BdObiF1 that surround the aromatic group rather than the β -hydroxyl group. We have revised this sentence (bottom of pg 11) to be more clear.

12. Figure 3b - should show full pocket and interactions; need better views of the substrate as well. “The side chain of Trp666, which adopts two alternate conformations, contributes to the hydrophobic pocket around the p-NO₂-phenyl ring” - show this. Distances should also be shown. Out of interest, what steps have been taken to “verify” the ligand density? Show difference density or a Polder map?

We have added a panel (now panel b) to Figure 3 that illustrates the specificity determining residues. In panel c, which now illustrates electron density, the density is an OMIT map, a standard approach for minimizing bias. The ligand occupancy is set to zero and the model is refined through a cycle of simulated annealing to remove bias. The alternate conformations of Trp666 are indeed illustrated in the figure in panel c, although not in panel b; this is described in the legend.

13. “Perhaps the BD-ObiF1 MLP promotes formation of the adenylylating conformation of the A domain” - speculation; either support or remove. In vitro assays with/ without MLP in trans could assist here.

Thank you for this suggestion. This Discussion has been largely re-written given our new experimental results.

14. MLP comparison of sequence should be done in a figure.

We have aligned several MLPs from structurally characterized systems and the Pf and BdObiF proteins and included this in Supplementary Figure 2. This facilitates the discussion of the MLP consensus sequence, which is described on pg 13.

15. “Therefore, there may be a subtle difference in MLP-A domain interactions depending upon whether the MLP domain is upstream or downstream of the A domain.” Same comment as above - support with evidence or remove the speculation. Trans interaction assays could help.

This Discussion has been largely re-written given our new experimental results.

16. MLP interactions in figure 3 c/d very hard to understand; figure needs to be reconsidered.

We have clarified in the figure legend that the A841 label refers to the alanine residue that lies under the transparent surface. We hope this makes the Figure more clear.

17. “This reduction in protein yield suggests that the MLP interaction with the ObiF1 adenylation domain does influence protein stability.” Could also simply be a folding issue before the MLP is even translated. Can’t say without more evidence that it is the loss of the MLP interaction that causes the loss in activity; it could be protein misfiling. Other mutants should be tested that are less extreme than Ala to Glu along with the idea to remove the MLP and include it in trans.

The goal of the mutation experiment was to strictly prevent the MLP from forming a viable interaction with the partner adenylation domain. The Ala to Glu mutation has been used by us (BR Miller, *JBC*, 2016) and others (DA Herbst, *JBC*, 2013) to block this interaction. We have also noted previously that a proline residue can substitute for the alanine (BR Miller). More conservative changes that allow the interaction would be less informative.

In the revised manuscript, we have supplemented this experiment with the production of a truncated ObiF1 protein that lacks the MLP domain or that is co-expressed with a separate MLP. All three variants (A841E, ObiF Δ MLP, and coexpressed ObiF delMLP + MLP, rbs1303/4) are less soluble than ObiF1 and show reduced activity. In our biochemical assay, the activity of the delMLP can be restored with added MLP. The A841E mutant requires a much higher level of MLP added back, illustrating the weaker affinity of the glutamate mutant.

We thank all three reviewers who raised this question in one form or another, prompting us to perform this more detailed series of experiments. Our new analysis is presented in Figure 4.

18. Why not reconstitute the whole system for the TE activity assays? Control cleavage of loaded starting material to establish conversion? Why the 24 hr timepoint?

Reconstitution of the full NRPS offers the advantage of monitoring direct product formation as opposed to build up of intermediates or consumption of substrates, which do not always correlate to product formation. We chose to quench reactions at 0.5, 3, and 24 hr to maximize signal to noise of the LCMS data. Earlier time points have higher error making mutant comparisons less informative. The 24 hr time point is a useful readout for the robustness of the ObiF1 mutants to determine if mutations impact stability and functionality over extended time periods. Additionally, for the substrate screens it provided extra information in the form of comparing product stabilities correlated to functionalization of the molecule.

19. Conversion of Cys to Ser? reference to Sherman’s work on TE characterisation during PKS biosynthesis again with a Cys-containing TE? Compare structure also to Schmeing/Chin’s new

trapped TE structure in Nature. Discussion of nocardicin TE domain would also seem appropriate.

We now include the Vlm2 TE domain from Schmeing/Chin in the comparison of Table S1 and also note in the discussion of the alternate triad conformation that the Vlm2 structure is like EntF and AB3403. We now also note that a similar enhanced reactivity was observed with the engineered PikTE mutant of Sherman (pg 27). [redacted]

20. Figure 5 - details of superposition / RMSD etc. are needed

The RMS displacement is now listed in Table S1, along with other comparisons.

21. R1190 - could it also be involved in PCP binding to the TE?

Superposition of our structure with 3TEJ, the structure of EntF TE-PCP complex from Bruner's lab shows that Arg1190 is at the opposite end of the TE domain lid from the PCP domain and is >20 Å from the PCP.

22. Figure 6 is very confusing and the data seems to be less indicative of general promiscuity but rather some scope for modification around the normal substrates.

We thank the reviewer for pointing this out. We updated our discussion of these experiments to reflect a preference for substrates with similarity to the native obafluorin substrates.

23. Why is DTT used for the substrate acceptance experiments and not GSSG? What are the differences if both are used on the same assay?

We explored several thiol capture agents including glutathione, cysteine, and DTT. We chose DTT for the substrate screens because it gives better resolution of products and provides information on the kinetics of the thiol capture reaction, transthioesterification, and S-to-O acyl transfer. DTT also confirmed the oxo ester is the thermodynamic product and thioester is the kinetic product. We have added some elements of this discussion to the main text.

24. "In most cases, the β -hydroxy- α -amino acids (2a-h) can be detected by LCMS (Figure 6b) indicating that ObiH is broad in substrate scope." I do not agree with this - the range of substrates used is small and many do not seem to work. Why not run this assay on its own as well to see what accumulates? Very hard to work out the selectivities of one enzyme amongst the activities of several enzymes. One option for some of the amino acid substrates would be to solve a co-crystal structure with these bound in the A-domain pocket to help justify some of the speculations made here. One could also make the A-domain assays competitive to see if the domain favours one substrate over the other.

See response to 22. We have performed more extensive studies of ObiH substrate scope that are the focus of a separate manuscript. We can confirm that ObiH is very broad in substrate scope, but does still prefer phenylacetaldehydes. ObiH alone is challenging to study because the reaction equilibrium favors starting materials over products ($K_{eq} \sim 0.2$). The coupled reaction with ObiF drives the equilibrium through ATP consumption.

Discussion:

25. There seems to me to be quite a bit of speculation in here; plus a narrative that links the work is somewhat lacking. Discussion of different substrate preferences - has this been tested? Nothing in here about the TE from nocardicin; nothing about other TE domains from NRPS systems that have been shown to act against different peptides. I do not think this TE domain is really a logic gate (or at least as I understand it) - a logic gate TE is the case when there are late stage modifications required from other enzymes prior to the TE becoming active; here the TE displays a different cleavage mechanism, but is not a logic gate. There is also no discussion of C-domain mediated b-lactam formation, which I would have expected.

Thank you for your suggestion. As noted above (Point 19) we plan to further compare different TE domains as well as different strategies for β -lactam/lactone formation in a subsequent publication. The reviewer is correct; Boddy has nicely defined TE domains as logic gates to ensure that appropriate chemical modifications have occurred prior to release. We no longer use this term, and have cleaned up this section. We have carefully reviewed the Discussion to remove some of the more speculative aspects, resulting in a cleaner description of our results.

Reviewer #2 (Remarks to the Author):

The paper by Kreitler et al describes the structure of the obafluorin forming NRPS ObiF1 that carries a MLP-domain as C-terminal domain. Overall, the unique biosynthesis of NRPS-derived beta-lactones and the widespread occurrence of such NRPS enzymes (see Scott et al) clearly justifies the paper, which is well written by experts in the field of NRPS structural biology. I really enjoyed reading the manuscript that contains a lot of interesting information regarding all the involved NRPS domains and gives a lot of new insights into this unusual type of NRPS biochemistry and NRPS in general.

However, I have a few comments that should be addressed by the authors:

26. Please correct and check *P. fluorescens* in the manuscript

We have made this correction.

27. I would suggest to add representations of both subtypes of ObiF enzymes and the two BGCs in both known producers to the SI

We have added this to the Supplementary Figure 1.

28. Page numbers are missing which make the paper difficult to review. But Bode was not an author in the paper describing the characterization of SlgN1 mentioned on p. 14? Either the citation or the authors must be corrected (the author I guess)

Apologies for this error. We have corrected this.

29. P.13: MLP consensus sequence: The similarity/differences are hard to see in the current form and I would suggest an alignment in the SI showing several sequences

We have added a sequence alignment with several MLP domains to Supplementary Figure 2.

30. A841E mutation shows a loss of ObiF1 activity but have the authors tried to cut the MLP domain off (and complement with it in trans?) This would be especially interesting with respect to the claims made in the discussion about the A domain specificity.

31. Can stand-alone MLPs substitute the cis MLP function?

Questions 30 and 31: Please see response above to question 17. We have performed these experiments and included this in the revised manuscript.

32. Along this line: what about the A domain-only specificity? Fig 6 shows the activity of a full ObiF1 protein, which might be different compared to the A domain.

Having successfully characterized the full NRPS module both structurally and functionally, we did not do any experiments to express the adenylation domain on its own. Indeed, a recent publication (A Degen, 2019, *Science Reports*) suggests that the activities of adenylation domains may differ when extracted from the full module. We were able to detect and now present in Figures S8 and S11 the benzoic acid acyl adenylates by LCMS and we included some comparison of acyl adenylate intermediates for ObiF1 mutants and substrate screens. These analyses did help inform substrate gating (and MLP dependence) by the ObiF1 and ObiF2 adenylation domains.

33. For me, it is clear that the domain ObiF1 domain organization is unique but it might be possible to get some knowledge especially about the MLP function with some minor experiments as suggested above.

Points 30-33. We have now performed and present several these experiments. Please see responses to point 13, 15, and 17, above.

34. Fig. 4: C and D are not good to read. Would a log scale be better? What is the detection limit of both compounds in this MS system?

We have updated Figure 4 using a log scale to better emphasize the impact of the mutations.

35. Fig. 6. The heatmap is a good representation of such a number of experiments that I like a lot. However, the ionization of the different compounds is not considered in such a representation and there might be huge differences especially for amine or halogen containing derivatives. Since the wt derivatives or closest relatives in both cases are produced best, it is in principle not necessary to compare the ionization but I would be careful using only signal intensity without making standard curves for the different compound classes or checking the ionization of the used acids. The amount of some derivatives might in fact be even much lower but suggesting a higher amount from a much better ionization (for example for af).

We agree and thank the reviewer for pointing this out.

We have compared peak heights in the optical absorbance spectra and peak heights in the extracted ion counts, which do suggest the compounds have similar ionization potential. For these studies, we were not trying to quantify the concentrations of products, we only strived to see if products could be formed (qualitative analysis) to gauge NRPS substrate selectivity and product formation.

36. Were other than benzoic acids tested as substrates in this assay? Is the A domain specificity of ObiF2 known? What about cinnamic acid or non-aromatic acids?

See response to point 32. We have added some discussion to the text about adenylation domain selectivity based on LCMS analysis of acyl adenylates.

Supplementary note:

37. Make all figures and tables understandable without the introduction of p.2 so please add all explanations to the figure/table captions (like NA etc)

We have revised figures and figure captions to improve clarity.

Supporting information:

38. Supp Fig 2: why is there such a large peak of 2a already at 0 min

The zero time point was analyzed immediately after addition of glutathione. This shows that reaction of obafluorin β -lactone with glutathione is quantitative and instantaneous; thus, validating our use of thiols as β -lactone trapping agents.

39. Supp Fig 2-4: I would suggest to use a slightly different color code not changing the color of the carbonyl group but for example showing H₂O in complete blue (comp. Supp Fig 4)

We appreciate the attention to this detail. We opted to keep these figures as is since they are in the supporting information.

40. Supp Fig 5: some derivatives seem to fluctuate in time like fa or ha. Have these experiments done multiple times or are these just minor differences?

We recorded time points at 0.5, 3, and 24 hr. Results for additional time points are provided in the supporting information document.

41. final general question: What can be said about the interaction between ObiD and ObiF that is essential for product formation? Are there COM or docking domains (especially at the N-terminus of ObiF1)? Is there a stable complex formed between ObiD and ObiF1F2? Has a pull down experiment been performed that might point to a stable interaction?

We had not previously considered this point. In both obafluorin NRPS architectures (*Pf* and *Bd*), all protein-protein interactions involve the free-standing carrier protein ObiD. These interactions are different than the traditional COM domain interactions that have been defined between catalytic domains that lie at the termini of multidomain NRPS proteins. In our experience, we have not been able to generate stable complexes between NRPS catalytic and carrier domains in the absence of specific chemical probes that serve to trap the interaction (Gulick and Aldrich, 2018, *NPR*). This is not surprising, given the required dynamics of the system that allows the carrier domains to dock temporarily to deliver the substrate for modification or off-loading.

Reviewer #3 (Remarks to the Author):

Gulick, Wencewicz and coworkers present structural and biochemical investigation of the obafluorin synthetase system. *Burkholderia diffusa* obafluorin synthetase subunit F1 has an unusual domain organization of C-A-T-TE-MLP, and has a TE domain that catalyzes a release step by cyclization to a β -lactone. The authors determined a crystal structure of *B. diffusa* ObiF1 and performed mutagenic and specificity analyses using a reconstituted obafluorin synthesis assay.

Overall, this is a very nice study that will add to the growing structural understanding of NRPS biology. The mutagenesis of the TE domain is nice, but without a substrate bound, it is difficult to fully understand why ObiF1 TE performs lactonization.

42. The most surprising finding is that in the structure of ObiF1, the MLP domain is bound to A1, implying that when the all proteins are present, the MLP domain will bind A1 and not AAr of ObiF2. This A1-MLP interaction is said to anchor the TE domain to the core of the module, which may be advantageous, but as is pointed out with AB3403, SrfAC and EntF examples, does not appear to be generally necessary. The authors test the importance of this interaction by an A841E mutation which should disrupt the ObiF1 A:MLP interaction, and which is shown to markedly decrease activity. However, this experiment cannot rule out that the A841E mutation is simply so destabilizing that the A domain is partially misfolded and therefore inactive only because of misfolding, not because lack of MLP binding. It would be very important and powerful to add some simple experiments with an ObiF1 C-A-T-TE construct, an ObiF1 MLP construct, and ObiF2 AAr: Does extracted ObiF1 MLP bind ObiF1 C-A-T-TE and/or ObiF2 AAr? Does ObiF1 MLP increases the adenylation activity of ObiF1 C-A-T-TE and/or ObiF2 AAr? (It would also be informative to know if ObiF2 bound ObiF1 in presence or absence of ObiD).

Thank you for this suggestion. As described above (point 17) and in the revised manuscript, additional experiments suggest that the functional MLP-Adenylation interaction are critical for the solubility (and activity) of the ObiF1 protein.

43. The result that D1254A generated approximately 25% as much β -lactone as wild type seems higher than expected. Can the authors comment on this and compare to available mutational data for other TE domains?

The reviewer is correct that several studies have shown an important role for the aspartic acid residue of the catalytic triad. [redacted]

44. I suggest shortening the “EntF-like triad configuration inhibits ObiF TE1 domain substrate loading” section. It is not surprising that simple reintroduction of ancestral residues into an evolutionarily diverged active site would not be sufficient to impart ancestral / full activity. Likewise, it is not uncommon that two mutations (G1173D/D1254A) are more deleterious than one (G1173D).

Thank you for your comments. We agree that there is likely a more complicated relationship between the EntF- and RifR-type catalytic triad conformations. Nonetheless, to our knowledge this analysis had not previously been performed and we felt that it would be interesting to test experimentally. Furthermore, we believe that the steric occlusion of the aspartic acid residue in the EntF-like position is supported by the G1173L mutant. The reviewer is correct that none of these results are shocking; however, we do feel that they give some insight into the underlying need for the novel catalytic triad conformation.

Minor

45. Abstract: “Nonribosomal Peptide Synthetase (NRPS)”/ The term nonribosomal peptide synthetase is not a proper noun and shouldn't be capitalized (It is not capitalized in the introduction.)

We have made this change.

46. “The NRPS antimicrobial β -lactone obafluorin is produced from a nonproteinogenic β -hydroxy amino acid.” Reword to clarify that the β -hydroxy amino acid is not the sole substrate as DHB is also a substrate.

We have revised as requested.

48. Introduction: “...differ significantly from known BGCs.” Perhaps rephrase to “from well-characterized BGCs.”

We have made this revision.

49. “ObiF1/F2 module” Is ObiF1/F2 a module?

We have replaced “ObiF1/F2 module” with “enzymes”

50. Results, page 11: “L-Thr is the simplest analog that mimics the β -OH-p-NO₂-homoPhe backbone stereochemistry and is not accepted as a substrate by the ObiF1 A domain. Therefore, structural features shared between Thr1 and the ObiF A domain are likely to be common among β -hydroxy acid adenylating enzymes.” I’m unsure why the inability of the ObiF1 A domain to accept L-Thr is indicative of common features.

The point we were making was that β -hydroxy amino acid activating domains may share features that recognize the β -hydroxy moiety. We have revised to clarify this sentence and further elaborate the interactions made between the protein and the β -hydroxy moiety.

51. Results: “In line with this hypothesis we observed a hydrogen bonding network of the catalytic triad at residues Cys1146, His1284, and Asp1254 (Figure 4b).” -Please show the hydrogen bonds in Figure 4b.

We have added dashed lines to panel 4b.

REVIEWERS' COMMENTS:

Reviewer #1 (Remarks to the Author):

I would like to commend the authors for their excellent work in responding to all of the reviewer's comments - I think that the extra data, re-worked manuscript and additional figures have really taken what was already a nice piece of work and made into a great one. All of my initial concerns/comments have been addressed to my satisfaction, and I am sure this paper will be well received in the broader biosynthesis community.

Dr Max Cryle

Reviewer #2 (Remarks to the Author):

The paper by Kreilter et al was nicely improved and I have no further comments except one: In the heat maps in Fig. 6 and 10 in the SI, the colors go from white (nothing) to dark blue (maximum). Why are there then back fields? Should they not be dark-blue?

This is a very good manuscript and I look forward to see it printed soon.

REVIEWERS' COMMENTS:

Reviewer #1 (Remarks to the Author):

I would like to commend the authors for their excellent work in responding to all of the reviewer's comments - I think that the extra data, re-worked manuscript and additional figures have really taken what was already a nice piece of work and made into a great one. All of my initial concerns/ comments have been addressed to my satisfaction, and I am sure this paper will be well received in the broader biosynthesis community.

Dr Max Cryle

Reviewer #2 (Remarks to the Author):

The paper by Kreilter et al was nicely improved and I have no further comments except one: In the heat maps in Fig. 6 and 10 in the SI, the colors go from white (nothing) to dark blue (maximum). Why are there then back fields? Should they not be dark-blue?

This is a very good manuscript and I look forward to see it printed soon.

We thank the reviewer for pointing out this concern. The heat maps that we have used do indeed range from white to dark blue. However, any peak that was above the maximum was colored black. This is now indicated in the legend for Figure 6 and Supplementary Figure 10.